# Integrative cross-omics and cross-context analysis elucidates molecular links underlying genetic effects on complex traits

Yihao Lu [1], Meritxell Oliva [1,2], Brandon L. Pierce [1], Jin Liu [3] ✉ & Lin S. Chen [1] ✉

Genetic effects on functionally related 'omic' traits often co-occur in relevant cellular contexts, such as tissues. Motivated by the multi-tissue methylation quantitative trait loci (mQTLs) and expression QTLs (eQTLs) analysis, we propose X-ING (Cross-INtegrative Genomics) for cross-omics and cross-context integrative analysis. X-ING takes as input multiple matrices of association statistics, each obtained from different omics data types across multiple cellular contexts. It models the latent binary association status of each statistic, captures the major association patterns among omics data types and contexts, and outputs the posterior mean and probability for each input statistic. X-ING enables the integration of effects from different omics data with varying effect distributions. In the multi-tissue cis-association analysis, X-ING shows improved detection and replication of mQTLs by integrating eQTL maps. In the trans-association analysis, X-ING reveals an enrichment of trans-associations in many disease/trait-relevant tissues.

Mapping genetic variants to their associated molecular 'omic' traits has emerged as an essential step in the functional annotation of genetic variants[1,2]. Rich resources of genetic and genomic data are made available for different molecular omic traits from various cellular contexts (e.g., tissues, cell types)[3–6]. It has been reported that expression quantitative trait loci (eQTLs) and methylation QTLs (mQTLs) often co-occur[7]. The detection of QTLs with cascading effects on multiple omics traits allows the study of functional variants and shared biological mechanisms, while also improving the power and precision for identifying disease/trait-relevant genes influenced by susceptibility loci[8,9]. To efficiently leverage the huge amount of existing data, summary statistics provide a convenient way with secured privacy in sharing data from a wide range of studies[1,10]. The integrative analysis of association summary statistics of genetic effects on multi-omics traits could collectively provide a comprehensive view and offer new insights into the dynamic mechanisms underlying human complex diseases and traits[11].

In a cross-omics integrative analysis, it is critical to consider the effect-operating cellular contexts. The genetic effects on omic traits depend on cellular contexts (e.g., tissues and cell types). Many disease-associated genetic and genomic effects are highly context-specific[5,12]. For example, studies of common-variant genetic results for schizophrenia (SCZ) showed concerted effects in certain cell types[13], while many different cell types play distinct functions in disease etiology. Moreover, most prior studies of genetic effects of molecular traits, i.e., QTL studies, are based on molecular trait measurements from bulk tissues, and those measurements are averaged across many functionally divergent cell types[5,14,15]. There could be multiple tissue types containing relevant cell types[16]. A joint analysis of many tissues, or broadly relevant cellular contexts and conditions, may reveal otherwise hidden and dynamic mechanisms underlying diseases of interest. Previous work integrating and leveraging summary-level data from a variety of cellular contexts[17], cell/tissue types[18,19], and conditions[20,21] show improved estimation and detection of disease/trait-relevant effects.

Recently, the enhancing GTEx (eGTEx) project sought to complement existing multi-tissue human transcriptome data with additional molecular traits, including DNA methylation (DNAm)[22]. DNAm

[1]Department of Public Health Sciences, The University of Chicago, Chicago, IL, USA. [2]Genomics Research Center, AbbVie, North Chicago, IL, USA. [3]School of Data Science, The Chinese University of Hong Kong-Shenzhen, Shenzhen, China. ✉e-mail: liujinlab@cuhk.edu.cn; lchen@health.bsd.uchicago.edu

data from 987 GTEx samples representing nine tissue types were made available to further characterize the relationships among inherited genetic variants, DNAm, gene expression and disease in a tissue-specific manner. Motivated by the multi-tissue mQTL and eQTL analysis, we propose X-ING (Cross-INtegrative Genomics) for cross-omics and cross-context integrative analysis. When integrating effects from multi-omics data, those effects do not share the same magnitudes and effect distributions. Yet there could still be linked or concerted patterns among true nonzero effects, e.g., co-occurring eQTLs and mQTLs[7]. The proposed X-ING method is built on a Bayesian hierarchical model capturing major patterns in the latent association status. A unique feature of X-ING is its ability to simultaneously account for omics-shared and omics-specific context/tissue-shared patterns in the analysis, while also allowing for effect heterogeneity and different levels of sparsity in each context and data type. Although X-ING is motivated by and primarily focuses on multi-tissue eQTL and mQTL analysis, it can be broadly applied to integrate multiple sets of summary statistics from different sources/domains to enhance cross-feature cross-context learning.

In this work, X-ING is used to enhance the power for detecting mQTLs by integrating eQTL statistics. Furthermore, we apply X-ING to examine multi-tissue trans-association effects. Our analysis reveals that associations identified by X-ING are enriched in many known disease/trait-relevant tissue types. Additionally, we illustrate how X-ING can integrate spatial differential expression statistics from spatial transcriptomic studies with multi-tissue eQTL statistics from GTEx. The integrative analysis provides new insights into spatially defined molecular mechanisms underlying diseases.

## Results

### An overview of the X-ING method for cross-omics and cross-context integrative analysis of QTL statistics

Recently, the GTEx consortium released the single-tissue mQTL summary statistics for nine selected tissue types from 424 GTEx participants. Due to the limited tissue-specific sample sizes, the detection of mQTLs is underpowered compared with existing eQTL maps. Moreover, a large majority of mQTLs lack functionality[23]. It has been reported that eQTLs often co-occur with mQTLs[7]. The joint analysis of associations of a genetic variant to a cis-gene and a cis-CpG site (a tested trio) could enhance the power and precision in detecting mQTLs, and facilitate the functional interpretations. We develop an integrative association method, X-ING, to jointly analyze multi-tissue mQTL and eQTL statistics, as illustrated in Fig. 1a–c. The X-ING method takes as input the association statistic matrices of $M$-tested units from $L$ types of omics studies, where each type of omics study has $K_\ell$ ($\ell = 1,...,L$) cellular contexts, i.e., $L$ matrices each of dimension $M \times K_\ell$. The outputs of X-ING are the posterior mean and the posterior probability of nonzero effect for each input statistic (Fig. 1d). The posterior probabilities allow for flexible inference. For example, we may identify mQTLs with multi-tissue effects (i.e., having effects in two or more DNAm tissues with posterior probabilities >80%), or mQTLs with co-occurring associations to cis-expression (i.e., also with effects in at least one expression tissue) at the 80% posterior probability cutoffs. One may also calculate false discovery rates (FDRs) based on the posterior probabilities[24,25].

In the motivating multi-tissue e/mQTL analysis ($L = 2$), a tested unit $i$ (=1,...,$M$) is a trio consisting of a single nucleotide polymorphism (SNP), a cis-gene, and a cis-CpG site from different human tissues, and $K_\ell$ is the number of tissue types in the $\ell$-th e/mQTL data. The input association summary statistics are $Z$-statistics obtained from single-tissue e/mQTL analysis. X-ING models the joint association patterns of $Z$-statistics (as $Z$-scores) for tested trios across omics data types and tissues. X-ING assumes those $Z$-scores from $K_\ell$ tissues, $z_{i\cdot,\ell}$, following a

multivariate normal distribution as follows

$$z_{i\cdot,\ell} \sim \mathcal{N}(\widetilde{z}_{i\cdot,\ell} \circ \gamma_{i\cdot,\ell}, \mathbf{R}_\ell), \tag{1}$$

where $\widetilde{z}_{i\cdot,\ell}$ is a vector of latent genetic association $Z$-scores (or effect sizes), with $\widetilde{z}_{ij,\ell} \sim \mathcal{N}(\widetilde{z}_{ij,\ell}|0, \sigma_{j,\ell}^2)$ in each of the $j$-th cellular context ($j = 1,...,K_\ell$), $\circ$ denotes the element-wise product of two vectors, $\gamma_{i\cdot,\ell} \in \mathbb{R}^{K_\ell}$ is a vector of latent binary association status, with one denoting the presence of a nonzero effect, and $\mathbf{R}_\ell \in \mathbb{R}^{K_\ell \times K_\ell}$ is a tissue-tissue correlation matrix (or covariance matrix if effect sizes instead of $Z$-scores being modeled) among all $K_\ell$ tissues due to potential sample overlap. The correlation matrix $\mathbf{R}_\ell$ due to sample overlapping (under the null, not of interest) is estimated a priori and is taken as known[8,21]. The latent association status indicator $\gamma_{ij,\ell}$ models sparsity in true nonzero effects. Similar modeling of latent indicators has been used in previous works to model the presence of nonzero $Z$-scores from multi-tissue eQTL analysis[21,26], and $Z$-scores from GWAS[27] among others. For different omics data types, effect size distributions are different. Some true nonzero effects are of opposite directions but are co-occurring. The joint modeling of latent indicators for nonzero effects captures shared effect co-occurrence patterns across data types despite different effect distributions – a major innovation of the proposed model. It should be noted that for inference purposes (i.e., detecting nonzero effects being the main goal), the modeling of $Z$-scores is similar to the modeling of effect sizes and their standard errors[21] and we use $Z$-scores as an illustration. Each of the $L$ latent $Z$-score matrices ($\widetilde{z}_{\cdot\cdot,\ell}$) captures the multivariate Gaussian distributions of latent $Z$-score values from multivariate contexts/tissues within each data type, while the $L$ latent binary matrices ($\gamma_{\cdot\cdot,\ell}$) further capture the major (low-rank) effect-sharing patterns across omics data types and contexts.

A key innovation of X-ING is that it models the patterns of latent binary association status together with effect sizes. This allows the integration of two or more statistic matrices from different data types of various effect distributions and arbitrary structures. In modeling of the latent binary association status $\gamma_{ij,\ell}$, X-ING links (with a logit link) it to a latent low-rank continuous matrix $\boldsymbol{U}_\ell \in \mathbb{R}^{M \times K_\ell}$ to capture the major effect-sharing patterns across omics and across contexts (Fig. 1d)[27],

$$\log \frac{p(\gamma_{ij,\ell} = 1 | \boldsymbol{U}_\ell, \boldsymbol{u}_{0,\ell})}{p(\gamma_{ij,\ell} = 0 | \boldsymbol{U}_\ell, \boldsymbol{u}_{0,\ell})} = U_{ij,\ell} + u_{0j,\ell}, \tag{2}$$

where $u_{0j,\ell}$ is the context-specific intercept, controlling the sparsity of nonzero effects in the $j$-th context of the $\ell$-th omics data. The latent matrix $\boldsymbol{U}_\ell$ captures the desired patterns in the data and modulates the latent probability of nonzero associations. When $\boldsymbol{U}_\ell = 0$, there is no borrowing information across contexts/data. Here we propose to capture omics-shared and data-specific context-shared patterns via latent low-rank approximated modulation matrices, $\boldsymbol{U}_\ell = \boldsymbol{U}_{\ell O} + \boldsymbol{U}_{\ell C}, \forall \ell \in \{1,...,L\}$, where $\boldsymbol{U}_{\ell O}$ represents the major (i.e., low-rank with rank $p_\ell$) omics-shared data structures for data $\ell$, and $\boldsymbol{U}_{\ell C}$ represents the major (rank $q_\ell$) data-specific context-shared structures for data $\ell$. With a computationally efficient expectation-maximization (EM) algorithm using variational inference, in each iteration, X-ING applies (generalized) canonical correlation analysis (CCA) on the logit-transformed latent probability matrices and retains the top $p_\ell$ canonical coefficients to obtain $\boldsymbol{U}_{\ell O}$. The number of retained components $p_\ell$ is determined using parallel analysis[28,29]. It then applies principal component analysis (PCA) on each residual matrix after subtracting the $\boldsymbol{U}_{\ell O}$ matrix to estimate the data-specific context-shared matrix $\boldsymbol{U}_{\ell C}$. The number of retained principal components is also determined using parallel analysis. The sequential estimation of $\boldsymbol{U}_{\ell O}$ and $\boldsymbol{U}_{\ell C}$ not only facilitates the computation

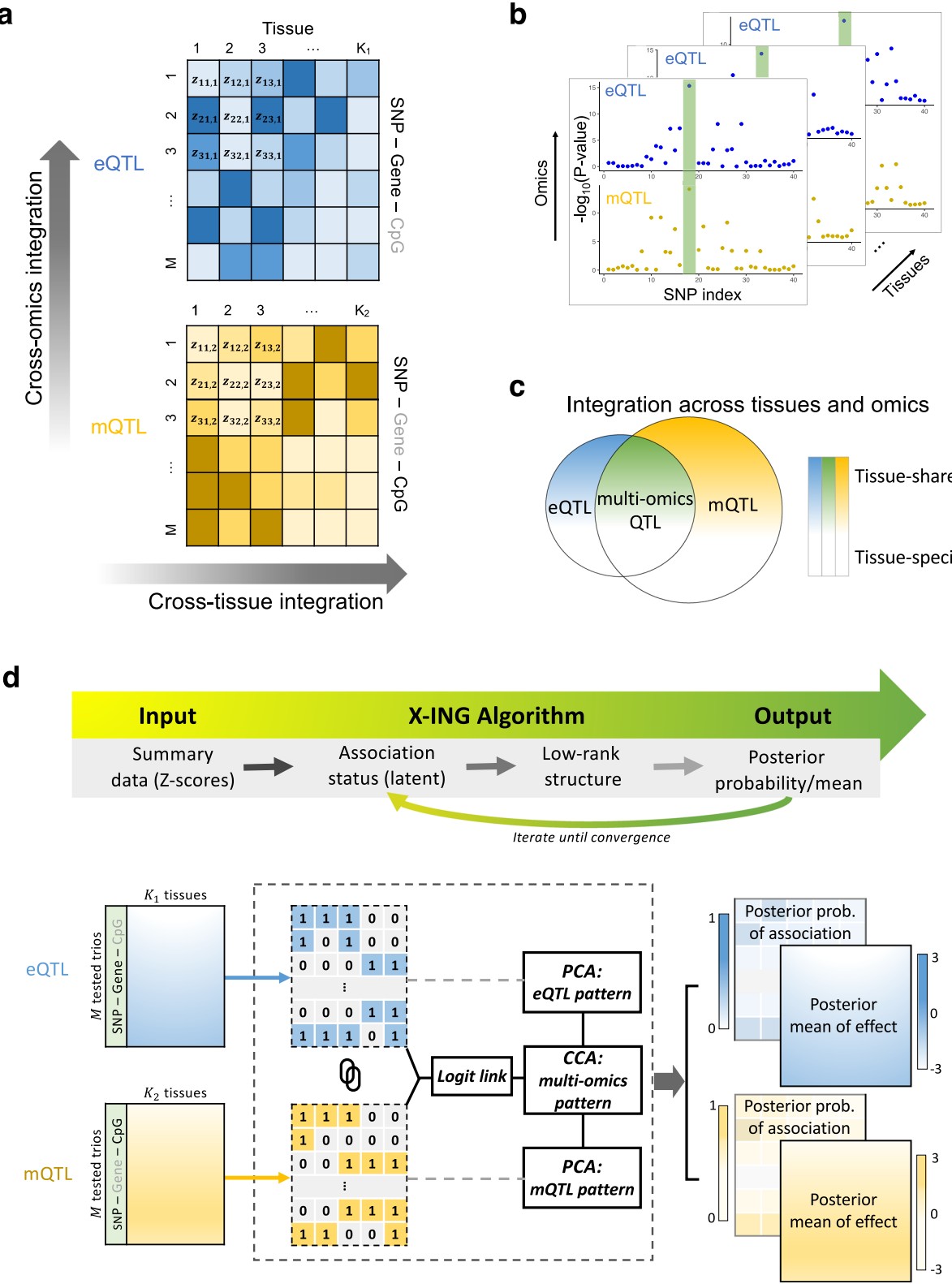

and also enhances the interpretability of the obtained CCs and PCs within the data context. See Supplementary Materials Algorithm 1 for details. The performance of X-ING is robust to the choices of $p_\ell$ and $q_\ell$ within a reasonable range. X-ING enables the integration of a broader class of test statistic matrices across different ($L$) data types while capturing the major structures of effects of similar nature from multiple ($K_\ell$) contexts (see "Methods" section).

## X-ING improves power by borrowing information across different data types and contexts

We conducted simulation studies to evaluate the performance of X-ING in comparison with existing methods in a variety of scenarios. We first simulated individual-level data across $L$ omics data, with $K_\ell$ contexts for each data $\ell$, under various combinations of between- and within-data correlations that represent variations in data-shared and

**Fig. 1 | Illustrations of the integrative analysis of multi-tissue e/mQTLs and the X-ING algorithm. a** An illustration of the multi-tissue e/mQTL integrative analysis. A total of $M$ trios are tested, each consisting a SNP, a cis-gene, and a cis-CpG site. eQTL data are from $K_1$ tissues and mQTL data are from $K_2$ tissues. **b** An illustration of existing multi-tissue QTL methods which analyze each omics data type separately or study the co-occurring patterns of associations tissue by tissue. The $P$-values in the figure only serve for illustration purposes. In practice, they can be obtained from QTL analyses. **c** The X-ING integrative analysis jointly analyzes multi-tissue eQTL and mQTL association statistics, borrows strengths across omics and tissue types, and captures association patterns that are omics-shared or tissue-shared. **d** An illustration of the X-ING algorithm via the multi-tissue e/mQTL analysis: X-ING takes as input $L = 2$ matrices of $Z$-statistics from eQTL and mQTL studies. It models the latent association status for each input statistic. Via a logit function, X-ING links the latent association status with a continuous modulation matrix for each data type. By performing CCA and PCA on the modulation matrices for e/mQTL data, X-ING captures the low-rank data-shared and data-specific major patterns. X-ING outputs the posterior probability (posterior prob. in the figure) and the posterior mean of association for each input statistic, accounting for those major patterns.

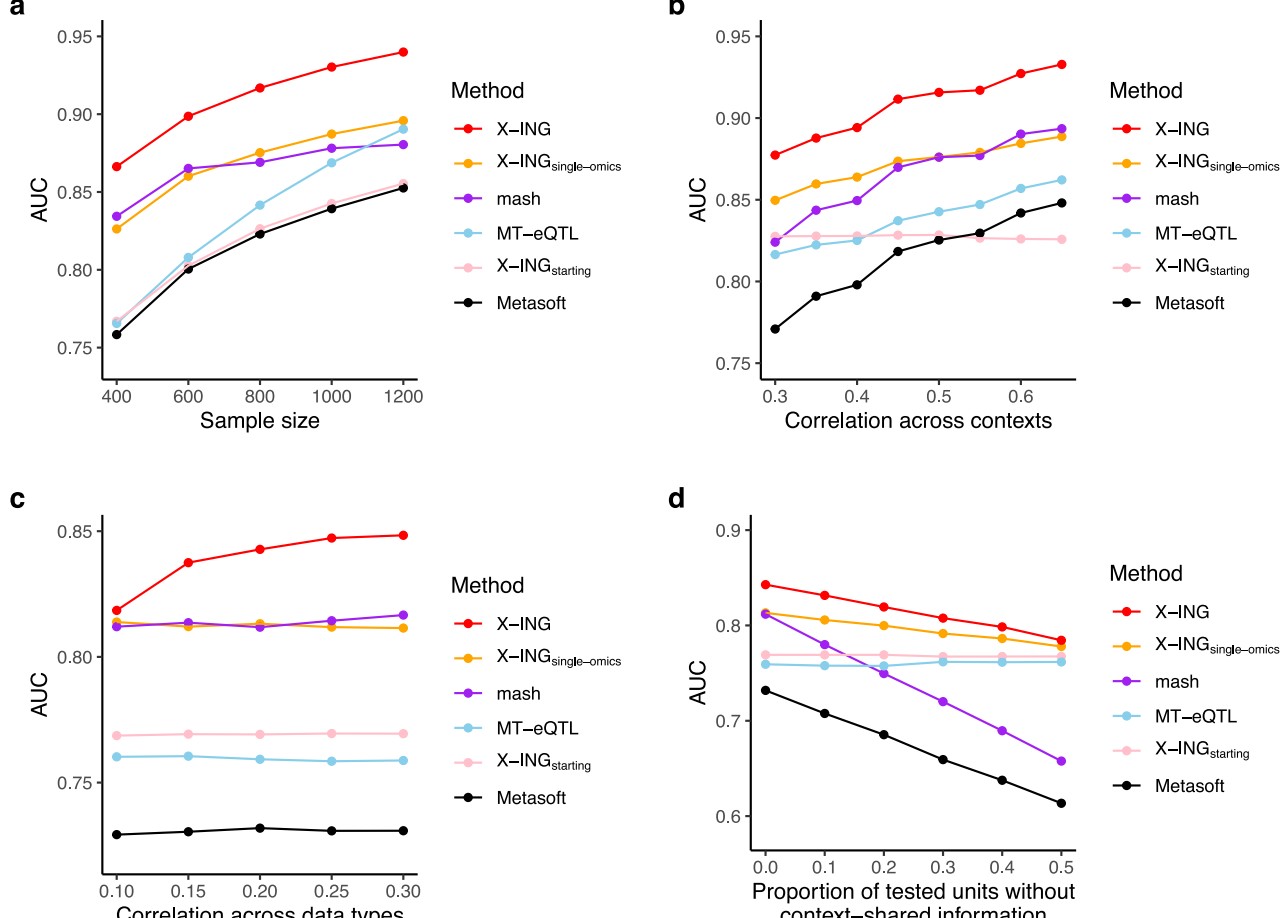

**Fig. 2 | Comparison of methods on simulated data. a** AUC for detecting the presence of nonzero effects on Data 1 with a sample size of Data 1 varying from 400 to 1200. We set $\rho_1 = \rho_2 = 0.5$, $r = 0.3$ and $\theta_1 = \theta_2 = 0.3$. **b** AUC on Data 1 with within-omics cross-context/tissue correlation, $\rho_1$, varying from 0.3 to 0.65. $N_1 = N_2 = 800$, $r = 0.2$ and $\theta_1 = \theta_2 = 0.3$. **c** AUC on Data 1 with cross-omics tissue-tissue correlation, $r$, varying from 0.1 to 0.3. $N_1 = N_2 = 400$, $\rho_1 = \rho_2 = 0.4$. **d** AUC on Data 1 with the proportion of tested units that do not have context-shared information varying from 0 to 0.5. $N_1 = N_2 = 400$, $\rho_1 = \rho_2 = 0.4$, $r = 0.2$ and $\theta_1 = \theta_2 = 0.3$.

-specific association patterns on $M$-tested units (see "Methods" section). Each set of data has a sample size of $N_\ell$, and the proportion of variation in the simulated response variables that can be explained by the predictors is $\theta_\ell$. By performing a simple linear regression on each response-predictor pair, we obtained $L$ matrices of $M \times K_\ell$ association statistics as input for the following analyses.

We compared X-ING with three existing methods: the multivariate adaptive shrinkage (mash)[21], multi-tissue eQTL (MT-eQTL)[26], and Metasoft[30]. Some of the existing methods were proposed for different purposes, and those methods can be adapted to the integrative analysis of summary data from multiple contexts for comparison purposes. We also included two variations of the X-ING models, X-ING$_{starting}$ and X-ING$_{single-omics}$, for comparison and illustration purposes. The X-ING$_{starting}$ is a starting model without the modeling of major data-shared patterns. It imposed the same prior for $\gamma_{\cdot,j,\ell}$ for all tested units in the $j$-th tissue from data $\ell$, i.e., no borrowing information across context and omics. The X-ING$_{single-omics}$ applies to only one omics data type. It considers shared patterns across contexts but does not allow the joint modeling of two or more omics data of different effect size distributions.

We first evaluated the selection performance (the predicted presence of nonzero effects) using the area under the curve (AUC) of the receiver operating characteristic (ROC) for all methods. We varied the sample size, within-data across-context correlation, between-data correlation, and proportion of tested units that do not have context-shared information. In all scenarios, X-ING outperformed other methods in terms of AUC. Not surprisingly, with increasing sample sizes, AUC of all six methods increased (Fig. 2a). All methods except for

X-ING_starting (no borrowing information) could gradually gain power as within-data correlation in Data 1 increased (Fig. 2b), suggesting multivariate integrative methods could improve power by borrowing information across contexts. The performance of the single-omics variation, X-ING_single−omics, was similar to that of the existing method for single-omics data, mash[21]. Mash and X-ING_single−omics both allow only cross-context/tissue integration but no borrowing information across data types. The former models effect sizes and the latter models the latent association status, and their performances are similar. When considering two omics data types, Fig. 2c showed that only X-ING could gain power as the between-data correlation increases. This is because X-ING borrows information across multi-omics data types. Allowing for multi-omics data integration is a major innovation of X-ING comparing with other methods. When the proportion of tested units without context-shared information increased, Fig. 2d showed that X-ING had higher AUC than other methods, and the AUC of mash and Metasoft reduced substantially.

Since many disease/trait-relevant effects are context-specific and cell-type-specific[12], we also evaluated the selection performance of detecting associations that were shared across two to five contexts (Table 1, see "Methods" section). X-ING achieved the highest overall and context-specific AUC compared to other methods. In the simulations of sparse associations (i.e., when there are many zero effects in the data), X-ING outperformed competing methods (Supplementary Fig. 1). When within- and between-data correlations were both zeros, all methods achieved similar performance as there was no shared information across contexts or data types (Supplementary Fig. 2). Supplementary Fig. 3 showed that by increasing the number of contexts, X-ING gained improved AUC due to more shared information across contexts. Moreover, X-ING performed better in data with varying percentages of variance explained by the predictors (Supplementary Fig. 4). We then evaluated the estimation of effect sizes using the root-mean-square error (RMSE) for both X-ING and mash, since only these two methods provided the posterior mean estimates. As sample size increases, both X-ING and mash gained improved RMSEs. The RMSE of X-ING was smaller than that of mash for data with different sample sizes (Supplementary Fig. 5).

The estimation of X-ING relies on parallelizable estimation of the posterior mean and probability for each test unit, as well as highly efficient algorithms (CCA and PCA) in capturing effect-sharing patterns. Supplementary Figs. 6 and 7 showed the comparison of the computation time of X-ING versus other methods. X-ING is computationally efficient. In addition, Supplementary Fig. 8 showed that X-ING was robust to weak correlations among predictors for tested units (i.e., when tested units have moderately dependent effects). We also showed that the performance of X-ING is robust to the choices of retained CCs and PCs in the CCA and PCA analysis within a certain range (Supplementary Fig. 9). When setting $U_\ell = 0$, the algorithm does not borrow information across data types nor tissue types. When setting $p_\ell$ or $q_\ell$ to full ranks, the model is over-parameterized. The low-rank approximation is useful in capturing major patterns in the data and borrowing information across omics data types and contexts.

## A multi-tissue cis-mQTL analysis integrating eQTL maps

The eGTEx project[22] generates DNA methylome data on subsets of GTEx samples from nine tissue types to study the genetic regulation of DNAm and expression across human tissues. Due to the limited DNAm tissue sample sizes, the detection of mQTLs is underpowered compared with existing eQTL maps. Moreover, a large majority of mQTLs lack functionality. To examine the genetic association patterns on DNAm together with expression variation in a tissue-specific manner while improving the functionality of the detected mQTLs, we applied X-ING to the cis-mQTL association statistics integrating eQTL maps ($L = 2$) generated on nine tissues representing $N = 367$ and 829 samples, respectively, from the GTEx project (v8)[5,23]. The list of tissues and tissue-specific sample sizes for mQTLs and eQTLs are provided in Supplementary Tables 1–2.

We obtained the sets of mQTLs from single-tissue mQTL analysis of eGTEx[23] including both lead and secondary mQTL variants for each CpG site within a 500KB cis-window size. We included 93,681 CpG sites and 159,186 unique mQTL variants, forming 204,220 unique SNP-CpG pairs. Each CpG site was assigned to a proximal gene with the nearest transcription start site (TSS)[31,32], forming 204,220 SNP-CpG-gene trios ($M = 204, 220$). We applied X-ING to the cis-eQTL and mQTL association statistics generated on 28 and nine tissues ($K_1 = 28$, $K_2 = 9$). In Supplementary Fig. 10, we showed that the major patterns/eigenvectors captured by X-ING can be interpreted as the surrogate variables for tissue-tissue dependence due to similar cell-type compositions.

At the 80% posterior probability cutoff (FDR[24,25] = 0.031), among the 204,220 analyzed SNP-CpG pairs, we identified a total of 143,801 pairs with nonzero mQTL effects in at least two tissues. Among those 143,801 SNP-CpG pairs (corresponding to 112,162 unique mQTL variants), 79,454 (58,158 unique variants) also exhibited nonzero association effect to its cis-gene expression in at least one tissue. In other words, more than half of the reported mQTLs also have nonzero associations to their cis-genes, suggesting joint genetic associations to both cis-DNAm and gene expression.

## Trans-association enrichment informs disease/trait-relevant tissues

To examine the trans-association enrichment patterns among diseases and traits, we conducted an integrative analysis using multi-tissue inter-chromosomal trans-e/mQTL association statistics ($L = 2$). We first selected 80 diseases/traits from a total of 216 diseases/traits and restricted the analysis to 40,466 GWAS SNPs associated ($P < 5 \times 10^{-8}$) with at least one of those 80 diseases/traits. Those SNPs were generally in weak linkage disequilibrium (LD) (Supplementary Fig. 11). We calculated the trans-eQTL association statistics in 28 GTEx tissues ($N \geq 73$; Supplementary Tables 1–2) and trans-mQTLs statistics in nine GTEx tissues (Supplementary Table 1) using FastQTL[33]. For each of the

**Table 1 | Comparison of methods on three sets of simulated data**

| Method | Overall AUC | | | AUC on associations true in two to five contexts | | |
|---|---|---|---|---|---|---|
| | $N_1 = 400$ | $N_2 = 800$ | $N_3 = 1200$ | $N_1 = 400$ | $N_2 = 800$ | $N_3 = 1200$ |
| X-ING | 0.823 | 0.881 | 0.904 | 0.856 | 0.914 | 0.933 |
| X-ING_single−omics | 0.797 | 0.850 | 0.873 | 0.819 | 0.871 | 0.891 |
| mash | 0.787 | 0.827 | 0.844 | 0.792 | 0.814 | 0.821 |
| MT-eQTL | 0.760 | 0.816 | 0.851 | 0.755 | 0.808 | 0.843 |
| X-ING_starting | 0.769 | 0.829 | 0.856 | 0.769 | 0.827 | 0.854 |
| Metasoft | 0.702 | 0.772 | 0.802 | 0.745 | 0.833 | 0.864 |

We evaluate the AUC of X-ING and competing methods in the analysis of three sets of statistics ($L = 3$). The overall AUC is calculated based on the true association status for all tested effects. We also compare the AUC for context-specific effects that have true nonzero effects in two to five cellular contexts/tissues.

80 selected diseases/traits, we applied X-ING to integrate the multi-tissue trans-association statistics for expression ($K_1 = 28$) and DNAm ($K_2 = 9$). At the 80% posterior probability cutoff, there were 644 to 15,490 SNP-gene-CpG site trios out of the examined trios for each selected disease/trait having nonzero trans-expression associations in at least one out of the 28 examined eQTL tissues, or having nonzero trans-methylation associations in at least one out of the nine examined mQTL tissues. We further identified trans-QTL hotspots with nonzero trans-effects on at least five genes/CpG sites. Further analysis showed that disease-associated hotspot regions explained more phenotypic variation compared with trait-associated ones (Supplementary Fig. 12).

We examined the enrichment of trans-expression associations across 28 tissue types by evaluating the scaled proportion of SNP-trans-gene pairs with trans-effects. Figure 3a showed the heatmap of the scaled proportion of SNP-trans-gene pairs identified with trans-associations for disease/trait-associated SNPs among the 28 tissues. As a comparison, Fig. 3b showed the corresponding heatmap of the scaled proportion for cis-expression associations. We observed a much stronger enrichment of trans-associations in many known disease/trait-relevant tissue types. For example, we identified the brain amygdala and prostate tissues as being enriched with trans-associations for Alzheimer's disease and prostate cancer, respectively[34,35]. The strong enrichment for trans-associations in many disease/trait-relevant tissues and the complementary patterns to cis highlight the potential of leveraging trans-expression associations together with cis in further identifying relevant tissues and cell types. It is consistent with the higher enrichment for heritability in regions surrounding genes with highly tissue-specific expression in disease-relevant tissue[36].

### Replication of cis- and trans-associations identified by X-ING

We evaluated the replication rates of SNP-CpG pairs with nonzero effects identified by X-ING. 119,401 out of 204,220 analyzed lead/secondary SNP-CpG pairs were also included in the replication data from FUSION (Finland-United States Investigation of NIDDM Genetics)

skeletal muscle study[37]. Among those 119,401 SNP-CpG pairs, 84,255 have nonzero effects in at least two tissues (posterior probability >0.8) in GTEx data. At the P-value threshold of $6 \times 10^{-7}$ (with Bonferroni correction)[38–40], 45.04% (53,780 out of 119,401) of the input SNP-CpG pairs were replicated in the FUSION data (without applying X-ING). In contrast, 55.79% (47,010 out of 84,255) of the SNP-CpG pairs identified by X-ING with multi-tissue effects (in two or more tissues) were replicated in FUSION. Moreover, we further categorized the examined mQTLs as (1) single-tissue mQTLs only, (2) single-tissue mQTLs with co-occurring expression associations, (3) multi-tissue mQTLs only, and (4) multi-tissue mQTLs with co-occurring expression associations. Table 2 shows the replication rates by tissue type for those four types of mQTLs. Not surprisingly, multi-tissue mQTLs are more likely to be replicated. It is worth noting that mQTLs with co-occurring expression associations have much higher replication rates than those without. Similar replication results were observed in GoDMC data (Table 2). The replication results demonstrate that by integrating multi-omics association studies and borrowing information across data types, X-ING improves the detection, replication, and functional interpretation of mQTLs.

We also evaluated the replication rates for trans-e/m associations identified by X-ING among the nine tissues with both expression and DNAm data, using eQTLGen[15] and GoDMC[39], respectively, as the replication studies. Among the 282 trans-expression associations identified by X-ING in the whole-blood tissue from GTEx, 56 of them (19.9%) were replicated at the 5% FDR level in the whole-blood-sample-based eQTLGen study. In contrast, the proportion of significant trans-expression associations among randomly selected SNP-trans-gene pairs in eQTLGen was 0.04%. Twenty-four of the 282 trans-eSNPs were trans-eSNPs in at least one another tissue besides blood and they were all replicated in eQTLGen. Moreover, 139 out of the 282 were also cis-eQTLs in at least one tissue, of which 44 were replicated in eQTLGen. The replication rate showed a 3.8-fold enrichment compared to trans-eSNPs that were not cis-eQTL ($P = 1 \times 10^{-6}$; two-sided Fisher's exact test). We observed similar patterns for trans-methylation associations.

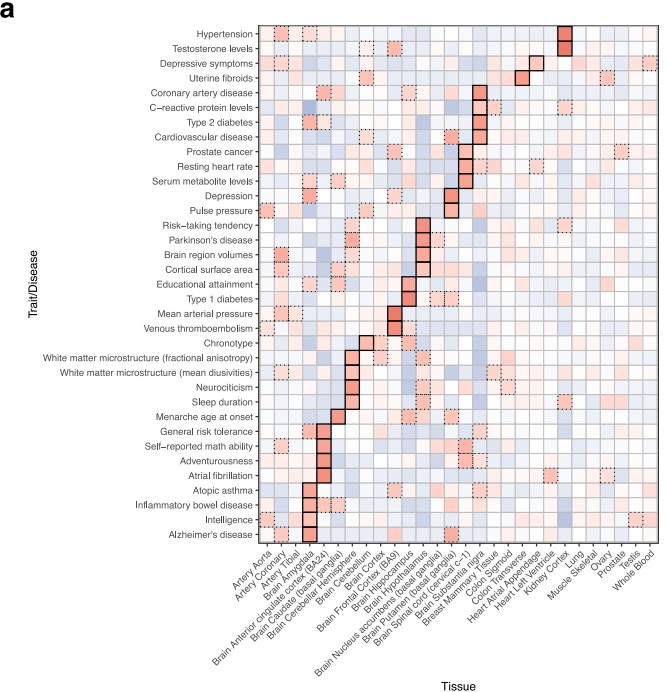

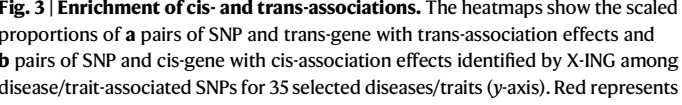

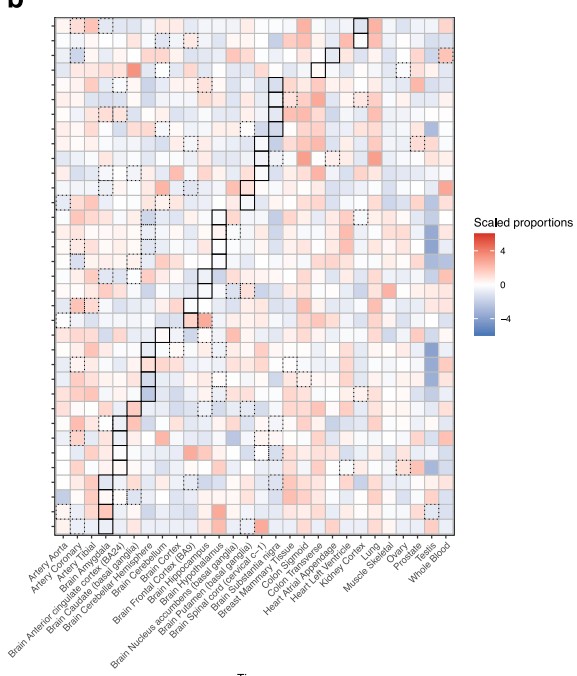

**Fig. 3 | Enrichment of cis- and trans-associations.** The heatmaps show the scaled proportions of **a** pairs of SNP and trans-gene with trans-association effects and **b** pairs of SNP and cis-gene with cis-association effects identified by X-ING among disease/trait-associated SNPs for 35 selected diseases/traits (y-axis). Red represents an enrichment and blue indicates a depletion of associations. We label the tissue with the highest level of trans-association enrichment for each disease/trait using solid lines. The tissue with the second and third highest enrichment in trans-association is labeled with dashed lines.

**Table 2 | Replication rates in the FUSION and GoDMC data for SNP-CpG associations identified by X-ING (posterior probability >0.8)**

| Replication study | Tissue of discovery | Sample size | Total # identified SNP-CpG pairs (PP > 0.8) | Type of effects | | | |
|---|---|---|---|---|---|---|---|
| | | | | Single-tissue, no expression association | Single-tissue, with expression association | Multi-tissue, no expression association | Multi-tissue, with expression association |
| FUSION | Breast mammary tissue | 49 | 13,870 | 0.08 (18/233) | 0.38 (18/47) | 0.74 (3259/4399) | 0.83 (7619/9191) |
| | Colon transverse | 189 | 83,803 | 0.17 (1013/6047) | 0.26 (955/3710) | 0.53 (16,532/31,140) | 0.63 (26,949/42,906) |
| | Kidney cortex | 47 | 13,690 | 0.39 (11/28) | 0.40 (6/15) | 0.64 (3489/5445) | 0.76 (6251/8202) |
| | Lung | 190 | 74,282 | 0.09 (427/4819) | 0.19 (464/2506) | 0.54 (14,957/27,733) | 0.63 (24,746/39,224) |
| | Muscle skeletal | 42 | 11,313 | 0.44 (24/54) | 0.75 (24/32) | 0.82 (3193/3881) | 0.91 (6672/7346) |
| | Ovary | 140 | 61,840 | 0.14 (922/6,431) | 0.24 (1067/4047) | 0.30 (11,658/21,215) | 0.40 (19,660/30,147) |
| | Prostate | 105 | 51,125 | 0.12 (258/1900) | 0.27 (273/906) | 0.32 (11,386/19,172) | 0.42 (20,081/29,147) |
| | Testis | 47 | 9631 | 0.11 (47/501) | 0.16 (28/179) | 0.33 (2182/3272) | 0.41 (4102/5679) |
| | Whole blood | 47 | 29,719 | 0.08 (34/442) | 0.30 (33/114) | 0.34 (6327/10,869) | 0.48 (12,638/18,294) |
| GoDMC | Breast mammary tissue | 49 | 24,583 | 0.07 (31/431) | 0.26 (20/78) | 0.31 (2474/8026) | 0.45 (7259/16,048) |
| | Colon transverse | 189 | 143,566 | 0.18 (1993/10,792) | 0.31 (1836/5960) | 0.31 (17,060/55,019) | 0.41 (29,614/71,795) |
| | Kidney cortex | 47 | 24,475 | 0.18 (10/55) | 0.56 (14/25) | 0.33 (3088/9412) | 0.42 (6255/14,983) |
| | Lung | 190 | 127,370 | 0.13 (1182/8820) | 0.26 (1003/3836) | 0.31 (15,359/48,930) | 0.42 (27,524/65,784) |
| | Muscle skeletal | 42 | 19,516 | 0.21 (16/77) | 0.29 (14/48) | 0.30 (2020/6837) | 0.36 (4572/12,554) |
| | Ovary | 140 | 105,088 | 0.13 (1476/11,079) | 0.23 (1482/6381) | 0.30 (10,895/36,923) | 0.40 (20,207/50,705) |
| | Prostate | 105 | 88,689 | 0.11 (388/3421) | 0.26 (394/1513) | 0.31 (10,751/34,421) | 0.42 (20,708/49,334) |
| | Testis | 47 | 18,767 | 0.09 (86/962) | 0.30 (96/323) | 0.33 (2123/6495) | 0.41 (4475/10,987) |
| | Whole blood | 47 | 52,558 | 0.09 (70/767) | 0.29 (60/205) | 0.34 (6762/19,717) | 0.47 (15,118/31,869) |

Proportions and numbers of SNP-CpG pairs identified in GTEx that are also significant in FUSION/GoDMC under $P$-value threshold of $6 \times 10^{-7}$ are listed. Those SNP-CpG pairs are divided into four groups based on the number of tissues and the presence of associations with cis-genes.

At the $P$-value threshold of $3 \times 10^{-9}$ (by Bonferroni correction)[39,41], 71 trans-methylation associations identified by X-ING were replicated in GoDMC, with a 7.4-fold enrichment ($P < 1 \times 10^{-16}$; two-sided Fisher's exact test) compared to randomly selected SNP-trans-CpG pairs. Among the 71 replicated trans-methylation associations, 44 of them were identified in at least two tissues in GTEx. Moreover, all 71 replicated trans-mSNPs were also cis-mQTLs in GTEx. Consistent with existing studies for cis-mediated mechanism of trans-associations[42–44], our results suggest that trans-e/m associations with joint trans- and cis-e/mQTL effects are more likely to be replicated.

**Tissue-sharing patterns of trans-association and cis-mediated trans-association effects**

To further characterize the tissue-sharing patterns of trans-effects mediated by cis-gene/CpG sites, we analyzed the cis- and trans-e/m association effects identified by X-ING in the nine shared tissues. Out of the 19,003 SNP-trans-gene pairs with trans-associations in at least one tissue, we first selected the pairs with cis-eQTLs. There were 7479 analyzed trios of SNP, cis-gene and trans-gene. Similarly, we selected 13,952 trios of SNP, cis-CpG site and trans-CpG site out of 14,433 SNP-trans-CpG pairs. For each trio, we estimated the indirect effect of a SNP on its trans-gene/CpG site via cis-gene/CpG site and the direct effect.

Figure 4 a showed that the total effects of 5934 (31.2%) SNP-trans-gene pairs and the indirect trans-expression association effects through cis-gene of 2160 (28.8%) SNP-cis-trans trios were shared by magnitude in at least two tissues. Here effects shared by magnitude refer to the effects with the same sign and within a factor of two of the strongest effect across tissues. Our findings are consistent with previous reports of the tissue-sharing patterns of indirect trans-association effects through cis-gene expression[42]. Moreover, we found similar effect-sharing patterns for trans-methylation associations (Fig. 4b). There were 4705 (32.6%) SNP-trans-CpG site pairs with shared trans-methylation association effects in at least two tissues. 3436 (24.7%) SNP-cis-trans trios shared similar indirect trans-mQTL effects via cis-CpG site in at least two tissues. Our results suggest that many trans-associations and cis-mediated trans-association effects are shared in some but not all tissue types. Proper multi-tissue analysis may enhance the power to detect them.

We plotted the negative log base 10 of the mediation $P$-values versus the percentage of reduction in trans-effects after accounting for putative cis-mediators (Supplementary Figs. 13–14). The percentage of reduction in trans-effects is also the ratio of indirect effect to total effect and is expected to be in the range of 0–1. A negative reduction in trans-effects with a significant mediation $P$-value suggests a potential false discovery of cis-mediation (see Supplementary Materials for details). Among trios with significant cis-mediated trans-effects (FDR < 0.05), trios identified as having multi-tissue trans-effects are less likely to have negative reductions in trans-effects, compared to trios identified as having single-tissue trans-effects in both e- and mQTL analyses. Those results suggest that multi-tissue analyses may reduce false discoveries of cis-mediated trans-association effects compared with single-tissue e/mQTL analyses.

**Integrating spatial transcriptomic data with multi-tissue eQTLs reveals spatially defined molecular links underlying SCZ genetics**

The X-ING method could also be applied to integrate broader and complementary sets of summary statistics to enhance cross-omics cross-feature learning. Here we apply X-ING to integrate differential expression statistics from spatial transcriptomic data with multi-tissue

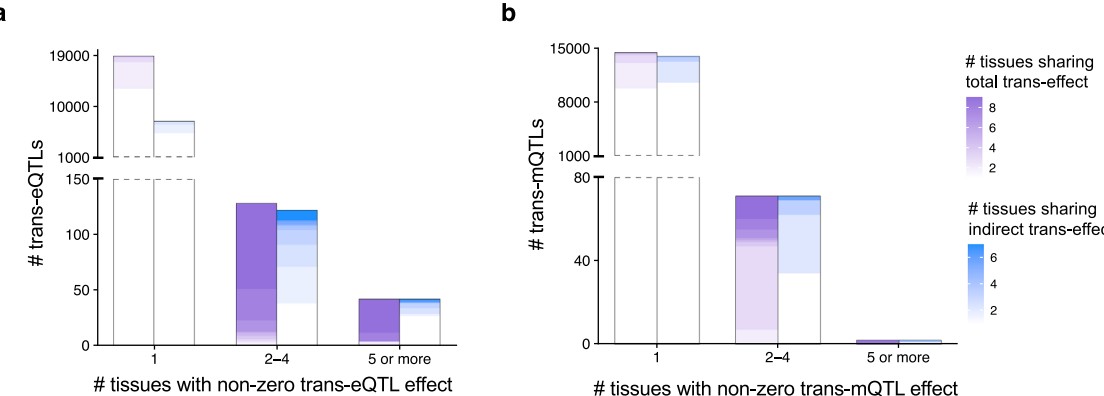

**Fig. 4 | Tissue-sharing patterns of trans-association effects and cis-mediated trans-association effects. a** Among the 19,003 selected SNP-trans-gene pairs, the total trans-eQTL effects of 5934 (31.2%; purple) pairs are shared in at least two tissues. Among the 7479 analyzed trios of SNP, cis-gene and trans-gene, the indirect trans-eQTL effects through cis-gene of 2160 (28.8%; blue) trios are shared in at least two tissues. **b** Among the 14,433 SNP-trans-CpG pairs, 4705 (32.6%; purple) pairs

have shared trans-mQTL effects in at least two tissues. Among the 13,952 trios of SNP, cis-CpG site, and trans-CpG site, 3436 (24.7%; blue) trios share similar indirect trans-mQTL effects via cis-CpG site in at least two tissues. Note that tissue-shared total/indirect effects refer to the effects with the same sign and within a factor of two of the strongest effect across tissues. The gradient color in each bar represents the number of tissues with shared effects.

eQTL statistics from GTEx. We detect the genes in cis-association with SCZ risk loci and also show laminar-specific expression (i.e., differential expression across different brain layers), accounting for the shared and data-specific patterns of the two sets of summary statistics[45]. Additionally, our results reveal the enrichment of laminar-specific expression of these genes in certain brain layers, offering valuable insights into spatially defined mechanisms underlying SCZ genetics.

To examine the laminar-specific variations for genes in cis with SCZ loci, we performed an integrative analysis of spatial differential transcriptomic statistics with multi-tissue eQTL statistics from GTEx brain tissues[5] ($L = 2$). We first performed spatial differential expression analysis[46] to obtain test statistics across six layers and white matter (WM) for each of the 12 samples ($K_1 = 7$) from Lieber Institute for Brain Development (LIBD) in human dorsolateral prefrontal cortex (DLPFC). Here we tested whether the expression of a gene in one layer differs from the other layers[46]. We obtained the set of brain eQTL statistics from 13 GTEx brain tissues ($K_2 = 13$). In total, we jointly analyzed over 1.6 million SNP-gene pairs ($M = 1.6$ million) matched in the spatial data from LIBD and the GTEx data. For SCZ, we included 8,962 SNP-gene pairs involving 527 SCZ risk SNPs and 3,184 genes in cis (1 MB) with a SCZ SNP. We performed X-ING analysis for each of the 12 samples. At the 90% posterior probability cutoff (FDR = 0.035), we identified genes differentially expressed in each specific layer in each sample and also associated with SCZ risk loci in at least two GTEx brain tissues. Among the 229 genes in cis-association with SCZ loci, a range of 9 to 41 genes exhibited laminar-specific expression for each pair of sample and brain layer. Further examination of these genes revealed that the laminar-specific expression of these SCZ-associated genes was enriched in layer 2 (L2; $P = 0.026$), layer 5 (L5; $P = 0.025$) and WM ($P = 0.070$) (Fig. 5). The significant enrichment in L2 and L5 were reported by existing studies[45], which demonstrated that SCZ risk genes in L2 and L5 showed decreased expression in SCZ patients. Here we also identified WM being enriched with genes that show spatially differential expression. By performing additional spatial registration of snRNA-seq datasets, Maynard et al.[45] reported preferential expression of oligodendrocyte subtypes in WM, where oligodendrocyte has been reported to contribute to neuropsychiatric disorders such as SCZ and autism spectrum disorder[47,48]. Supplementary Fig. 15 showed the significance of layer enrichment for differentially expressed genes associated with autism spectrum disorder (ASD) risk loci in at least two GTEx brain tissues. There was an enrichment of differentially

expressed genes in L2 ($P = 0.009$), L5 ($P = 0.030$), L6 ($P = 0.028$), and WM ($P = 0.020$) for cis-genes associated with ASD risk loci.

## Discussion

In this work, we propose X-ING as a general framework for the cross-integration of summary statistics from multi-omics data each with multiple cellular contexts. X-ING takes as input the summary statistic matrices from $L$ data types and models each input statistic as a product of Gaussian and latent binary association status. The modeling of $L$ latent binary matrices allows the cross-integration of different data types of different effect distributions, and X-ING captures omics-shared and context-shared association patterns. This is a major innovation compared with existing multi-context/tissue methods analyzing only one data type at a time. Additionally, X-ING allows for different levels of sparsity in each context, potential sample overlapping, and effect heterogeneity. With simulation studies, we demonstrate that X-ING improves the estimation of association probabilities and effect sizes in various simulated settings by borrowing strengths across different data types and contexts.

We applied X-ING to detect multi-tissue cis-mQTLs integrating eQTL maps, with a focus on cis-mQTLs with co-occurring associations in other omics data and contexts. We examined trans-e/m association patterns across multiple tissues from GTEx, with a focus on the disease/trait-associated SNPs. The cis-mQTLs and trans-e/m associations identified by X-ING were replicable, especially for those with effects identified in multiple tissues or omics data types. The enrichment of trans-associations in tissue types is informative in suggesting the disease/trait relevance of tissues. We also characterized the tissue-sharing patterns of total effects and indirect effects of trans-association through cis-mediators. In another analysis, we illustrate the broader application of X-ING by integrating spatially differential expression statistics from spatial transcriptomic data with multi-tissue eQTL statistics from 13 GTEx brain tissues. We highlighted the spatial heterogeneity in expression variation of many SCZ risk-associated genes and provided new insights into studying the spatially defined mechanisms underlying SCZ genetics.

There are some limitations and caveats of current work. First, the detected joint associations across multi-omics data or in multiple cellular contexts are not evidence of causation. X-ING does not perform colocalization analysis. Though the findings of X-ING may provide insights of potential connected relationships and mechanisms, it should be interpreted as associations. Second, the cross-integrative methods

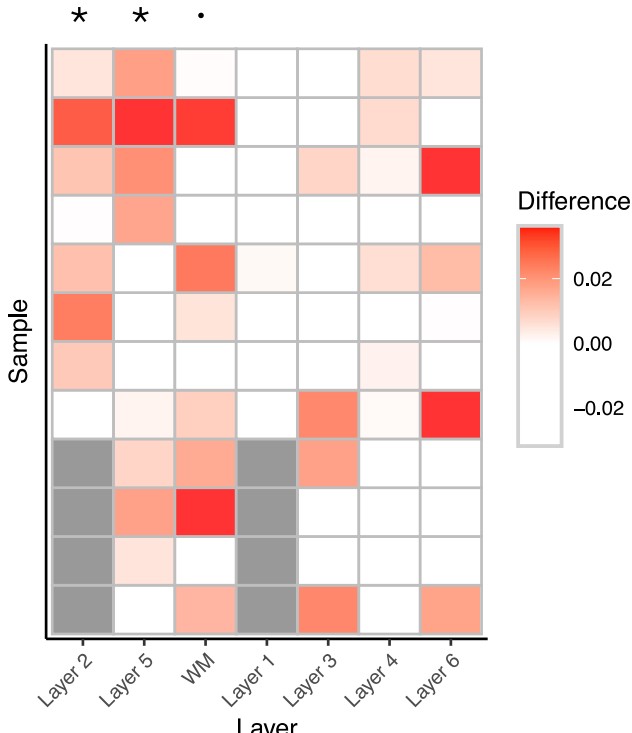

**Fig. 5 | Laminar-specific expression of SCZ-associated genes.** The heatmap shows the enrichment of layer-specific differentially expressed genes among disease risk-associated genes, compared with the proportions of layer-specific differentially expressed genes across the genome. The color in each cell indicates the difference between the two proportions for each sample and layer. Red represents an enrichment of differentially expressed genes in a specific sample and layer, and white represents a depletion of differentially expressed genes. Gray cells indicate missing values (no distinct layer information). There is an enrichment of differentially expressed genes in layer 2 ($P = 0.026$) with $t(7) = 2.34$, layer 5 ($P = 0.025$) with $t(11) = 2.20$, and white matter ($P = 0.070$) with $t(11) = 1.59$ for cis-genes associated with SCZ risk loci. $P$-values were calculated by paired one-sided $t$-test without multiple testing adjustment, based on the listed t-statistics and degrees of freedom.

are not powerful in detecting effects and associations that are specific to only one omics data type in only one context. For those omics- and context-specific effects, existing multivariate methods may not improve power and context-specific sample size is still the major limiting factor. Third, X-ING treats the $M$-tested units as independent in the estimation. When analyzing published disease/trait-associated SNPs or single-tissue QTLs, most of them are uncorrelated or in weak LD. In general, we recommend applying X-ING to tested units with at most moderate dependence. Last but not least, X-ING does not allow missingness in the input statistics, and a naive imputation may facilitate the analysis but may induce biases if there is substantial missingness.

In future work, X-ING can be improved with a more efficient and selective data integration, when the number of available sets of summary statistics is high, e.g., $L \geq 5$ and some $K_\ell \geq 50$. Another potential area of future development is the integration of association statistics with mediation and causal estimates from multiple studies to reduce confounding and spurious associations.

## Methods
### A starting Bayesian model without the modeling of shared data patterns
Assuming independence among all $M$-tested units/trios, for each trio $i$ ($i = 1, ..., M$) in omics data $\ell$, we assume its $Z$-scores from $K_\ell$ tissues, $\mathbf{z}_{i,\ell}$, following a multivariate normal distribution as Eqn. (1). We further assume that each latent genetic association $Z$-score $\tilde{z}_{ij,\ell}$ takes a

Gaussian prior, $\mathcal{N}(\tilde{z}_{ij,\ell}|0,\sigma_{j,\ell}^2)$, and each $\gamma_{ij,\ell}$ takes a Bernoulli distribution with success probability $\pi_{j,\ell}$, which controls the sparsity of non-zero effects in tissue $j$ for data $\ell$. This specification is equivalent to assuming a spike-slab prior for the product, denoted as $\boldsymbol{\eta}_{i\cdot,\ell}$, of $\tilde{\mathbf{z}}_{i\cdot,\ell}$ and $\boldsymbol{\gamma}_{i\cdot,\ell}$[49,50], with each $\eta_{ij,\ell}$ distributed as

$$\eta_{ij,\ell} \sim \begin{cases} \mathcal{N}\left(\eta_{ij,\ell}|0,\sigma_{j,\ell}^2\right), & \text{if } \gamma_{ij,\ell} = 1, \\ \delta_0(\eta_{ij,\ell}), & \text{if } \gamma_{ij,\ell} = 0. \end{cases} \quad (3)$$

where $\delta_0$ is a Dirac delta function at zero. To promote computational efficiency, we utilize the specification in Eqn. (1) (see Supplementary Materials). The complete-data likelihood can be written as

$$p(\mathbf{z},\tilde{\mathbf{z}},\boldsymbol{\gamma};\Theta_1) = \prod_{\ell=1}^{L}\left(\prod_{i=1}^{M}p(\mathbf{z}_{i,\ell}|\tilde{\mathbf{z}}_{i,\ell},\boldsymbol{\gamma}_{i,\ell}) \cdot \prod_{i=1}^{M}\prod_{j=1}^{K_\ell}p\left(\tilde{z}_{ij,\ell}|\sigma_{j,\ell}^2\right)p\left(\gamma_{ij,\ell}|\pi_{j,\ell}\right)\right),$$
$$= \prod_{\ell=1}^{L}\left(\prod_{i=1}^{M}\mathcal{N}(\mathbf{z}_{i,\ell}|\tilde{\mathbf{z}}_{i,\ell}\circ\boldsymbol{\gamma}_{i,\ell},\mathbf{R}_\ell) \cdot \prod_{i=1}^{M}\prod_{j=1}^{K_\ell}\mathcal{N}\left(\tilde{z}_{ij,\ell}|0,\sigma_{j,\ell}^2\right)\cdot\pi_{j,\ell}^{\gamma_{ij,\ell}}\left(1-\pi_{j,\ell}\right)^{1-\gamma_{ij,\ell}}\right),$$
$$(4)$$

where $\Theta_1 = \{\mathbf{R}_\ell,\sigma_{j,\ell}^2,\pi_{j,\ell},\ell=1,\ldots,L,j=1,\ldots,K_\ell\}$ is the collection of model parameters and $\circ$ denotes element-wise product of two vectors. In practice, the tissue-tissue correlation matrix $\mathbf{R}_\ell$ due to sample overlap is often pre-estimated and treated as known[21]. Existing literature often estimates it using a subset of the input $Z$-statistics that are likely to be from the null distributions (for example, using only SNPs with $|Z| < 5$ in all tissues to estimate the tissue-tissue correlation matrix)[8,21]. Note that in the starting model (4) without modeling shared data patterns, the prior for $\boldsymbol{\gamma}_{\cdot j,\ell}$ is the same for all tested units in the $j$-th tissue from data $\ell$.

To obtain the estimates of parameters in model (4), we need to compute the conditional density of latent variables given the observed $Z$-scores,

$$p(\tilde{\mathbf{z}},\boldsymbol{\gamma}|\mathbf{z};\Theta_1) = \frac{p(\mathbf{z},\tilde{\mathbf{z}},\boldsymbol{\gamma};\Theta_1)}{p(\mathbf{z};\Theta_1)}, \quad (5)$$

where $\mathbf{z}$, $\tilde{\mathbf{z}}$ and $\boldsymbol{\gamma}$ are the collections of $\mathbf{z}_\ell$'s, $\tilde{\mathbf{z}}_\ell$'s and $\boldsymbol{\gamma}_\ell$'s, respectively. To facilitate computation, we could apply an empirical Bayes approach to estimate the conditional density with specification of distributions for hyperparameters. EM algorithms are usually used to obtain the estimates for models with latent variables. Here, the difficulty of employing standard EM algorithms comes from two folds. First, due to the curse of dimensionality, it would be computationally intensive to evaluate the expectation of high-dimensional latent variables. Second, it would be computationally intractable to integrate out the latent variables with spike-slab priors. To efficiently estimate the parameters in model (4), we use an EM algorithm with variational inference (see Supplementary Materials).

### A Bayesian hierarchical model for the X-ING method
When jointly analyzing summary statistics from multi-omics data each with multivariate cellular contexts, some association patterns are shared between omics data types. For example, eQTLs and mQTLs often co-occur. Some cellular contexts are more correlated than others. The proposed cross-integrative genomics method, X-ING, accounts for those omics-shared and context-shared association patterns. In contrast to existing single-omics methods modeling effect sizes from multiple contexts, the proposed X-ING method jointly analyzes $L$ matrices of $Z$-statistics and models the latent binary association status, $\{\boldsymbol{\gamma}_\ell\}$, to facilitate the modeling of effect co-occurring patterns from multi-omics data of different nature and distributions. X-ING enables a broader class of integrative analyses across different ($L$) data types while also allowing the integration of statistics of similar nature across different ($K_\ell$) contexts – a major advantage of

X-ING. We develop an EM algorithm with variational inference (see Supplementary Materials) for the model. In each E-step, we evaluate the posterior distribution of latent variables and obtain the variational parameters. In the M-step, we extract the common features shared among matrices of logit-transformed posterior probabilities of latent binary status by performing a CCA and PCA.

In detail, we build a Bayesian hierarchical framework to model the structured major patterns in the latent association status, $\gamma_{ij,\ell}$. To promote sparsity and model major data patterns across multiple omics and contexts, we modulate the prior probability of the latent status and link it with a latent low-rank matrix $\boldsymbol{U}_\ell \in \mathbb{R}^{M \times K_\ell}$ that captures omics-shared and context-shared major patterns via a logit link, as Eqn. (2).

Via the above modeling, our method efficiently allows the prior probability being specific for each tested unit (pair, trio, etc.) without over-parametrization, $p(\gamma_{ij,\ell} = 1 | \boldsymbol{U}_\ell, \boldsymbol{u}_{0,\ell}) = \pi_{ij,\ell}$. Moreover, the joint estimation of low-rank matrix $\boldsymbol{U}_\ell$ across $L$ data types also allows us to capture the shared information across different data types with an unknown extent of information sharing. Combining the prior probability in (2) and the Gaussian prior for $\tilde{\boldsymbol{z}}$ with the multivariate normal distribution for $Z$-scores in (1), the complete-data likelihood for X-ING can be written as

$$p(\boldsymbol{z},\tilde{\boldsymbol{z}},\boldsymbol{\gamma}) = \prod_{\ell=1}^{L}\left(\prod_{i=1}^{M}\mathcal{N}(\boldsymbol{z}_{i,\ell}|\tilde{\boldsymbol{z}}_{i,\ell}\circ\boldsymbol{\gamma}_{i,\ell},\boldsymbol{R}_\ell)\cdot\prod_{i=1}^{M}\prod_{j=1}^{K_\ell}\mathcal{N}\left(\tilde{z}_{ij,\ell}|0,\sigma_{j,\ell}^2\right)\cdot\pi_{ij,\ell}^{\gamma_{ij,\ell}}\left(1-\pi_{ij,\ell}\right)^{1-\gamma_{ij,\ell}}\right).$$

(6)

Comparing with the starting model (4), the X-ING model (6) allows each tested unit to have a specific prior that is modulated via a low-rank term, $U_{ij,\ell}$, based on a logit transformation of the latent association indicator, $\gamma_{ij,\ell}$.

When considering an $M \times K_\ell$ matrix of statistics from a single-omics data ($L=1$), there exists no data-shared structure and we may regularize the rank of $\boldsymbol{U}_\ell$ using the nuclear norm[27]. If association pattern sharing is limited across $K_\ell$ tissues or when a larger regularization is imposed, the low-rank matrix $\boldsymbol{U}_\ell$ may become a zero matrix and the prior model (2) reduces to a Bernoulli random variable with a shared probability parameter as in model (4), i.e., $p(\gamma_{ij,\ell} = 1 | \boldsymbol{u}_{0,\ell}) = \pi_{j,\ell}$, indicating only tissue-specific sparse priors being imposed. When jointly estimating the low-rank matrices across $L$ data types, we could estimate the latent low-rank matrix $\boldsymbol{U}_\ell$ as $\boldsymbol{U}_\ell = \boldsymbol{U}_{\ell O} + \boldsymbol{U}_{\ell C}$, where $\boldsymbol{U}_{\ell O}$ captures the omics-shared major patterns across $L$ omics data types, and $\boldsymbol{U}_{\ell C}$ captures data-specific context-shared patterns within-data type $\ell$ across $K_\ell$ contexts. In detail, we could extract the common latent features, $\boldsymbol{U}_{\ell O}$'s, shared among two or more omics data types by performing a CCA or a generalized canonical correlation analysis (GCCA) on the logit-transformed latent probability matrices using the R package RGCCA[51]. The number of retained components $p_\ell$ is determined using parallel analysis[28,29]. Additionally, X-ING further models the major sharing patterns, $\boldsymbol{U}_{\ell C}$, across tissues/contexts within each omics data type by performing PCA on each residual matrix after subtracting the $\boldsymbol{U}_{\ell O}$ matrix. The number of retained principal components is also determined using parallel analysis. The modeling of omics-shared and data-specific patterns may not be uniquely identifiable and they do not need to be. That is, $\boldsymbol{U}_{\ell O}$ and $\boldsymbol{U}_{\ell C}$ do not need to be orthogonal, and their sum $\boldsymbol{U}_\ell$ could still be a good approximation of the logit-transformed probabilities of latent indicators. X-ING performs CCA and PCA sequentially within each iteration, and simultaneously accounts for omics-shared and omics-specific context-shared association patterns across data types and contexts. It outputs the posterior mean and probability of nonzero for each input statistic. Additionally, it provides the eigenvectors from PCA and the canonical coefficients from CCA at the final iteration, and these outputs may facilitate the interpretations of the major patterns in the data.

## Generation of summary statistics in the simulation studies

We generated $L$ matrices of $M \times K_\ell$ association summary statistics using simulated individual-level data. We first simulated predictor variables $\boldsymbol{X}_\ell$, with $M$ omics-specific predictors and a sample size of $N_\ell$. Each element of $\boldsymbol{X}_\ell$ was generated independently from a standard normal distribution. Here, we simulated binary association status, $\gamma_{ij,\ell}$ with given correlation structure using the R package bindata. By simulating effect size, $\beta_{ij,\ell} \sim \mathcal{N}(0,\sigma_\ell^2)$, for each data type $\ell$, we then considered the following equation to generate response variables,

$$\boldsymbol{y}_{j,\ell} = \boldsymbol{X}_\ell(\boldsymbol{\beta}_{j,\ell} \circ \boldsymbol{\gamma}_{j,\ell}) + \boldsymbol{\epsilon}_{j,\ell},$$

(7)

where $\boldsymbol{\epsilon}_{j,\ell}$ was the error term following $\mathcal{N}(0,\sigma_\epsilon^2)$. In the simulation studies, we controlled the proportion of variation in the response variable, $\boldsymbol{y}_{j,\ell}$, that can be explained by the predictors, $\theta_{j,\ell} = \frac{\text{var}(\boldsymbol{X}_\ell(\boldsymbol{\beta}_{j,\ell}\circ\boldsymbol{\gamma}_{j,\ell}))}{\text{var}(\boldsymbol{y}_{j,\ell})}$, where we assumed the same $\theta_\ell$ for all $\theta_{j,\ell}$'s in data $\ell$.

In the simulation of two omics data types, i.e., $L=2$, each row of the corresponding association status matrices $\boldsymbol{\gamma}_1 \in \mathbb{R}^{M \times K_1}$ and $\boldsymbol{\gamma}_2 \in \mathbb{R}^{M \times K_2}$ was jointly simulated[52] with the correlation matrix $\boldsymbol{\Omega}$ being:

$$\boldsymbol{\Omega} = \begin{pmatrix} \boldsymbol{W}_1 & \boldsymbol{C} \\ \boldsymbol{C}^\top & \boldsymbol{W}_2 \end{pmatrix},$$

(8)

and probability of being 1 as $\tau_\ell$ ($\ell = 1, 2$). Here $\boldsymbol{W}_\ell \in \mathbb{R}^{K_\ell \times K_\ell}$ ($\ell = 1, 2$) was the within-data correlation structure across contexts for the $\ell$-th data type, and $\boldsymbol{C} \in \mathbb{R}^{K_1 \times K_2}$ was the between-data correlation matrix. Within each omics data type $\ell$, all $K_\ell$ contexts can be partitioned into two groups, with $K_\ell^{(1)}$ and $K_\ell^{(2)}$ contexts, respectively, where $K_\ell^{(1)} + K_\ell^{(2)} = K_\ell$. We also considered nonzero within-group context-context correlations for each group of contexts. The correlation matrices $\boldsymbol{W}_\ell$'s and $\boldsymbol{C}$ were specified as

$$\boldsymbol{W}_\ell = \begin{pmatrix} (1-\rho_\ell)\cdot\boldsymbol{I}_{K_\ell^{(1)}}+\rho_\ell\cdot\boldsymbol{1}_{K_\ell^{(1)}}\boldsymbol{1}_{K_\ell^{(1)}}^\top & \boldsymbol{0} \\ \boldsymbol{0} & (1-\rho_\ell)\cdot\boldsymbol{I}_{K_\ell^{(2)}}+\rho_\ell\cdot\boldsymbol{1}_{K_\ell^{(2)}}\boldsymbol{1}_{K_\ell^{(2)}}^\top \end{pmatrix},$$

$$\boldsymbol{C} = \begin{pmatrix} r\cdot\boldsymbol{1}_{K_1^{(1)}}\boldsymbol{1}_{K_2^{(1)}}^\top & \boldsymbol{0} \\ \boldsymbol{0} & r\cdot\boldsymbol{1}_{K_1^{(2)}}\boldsymbol{1}_{K_2^{(2)}}^\top \end{pmatrix},$$

(9)

where $\boldsymbol{1}_{K^{(1)}}$ and $\boldsymbol{1}_{K^{(2)}}$ were column vectors of ones with length $K_\ell^{(1)}$ and $K_\ell^{(2)}$, respectively, $\rho_\ell$ was the pair-wise correlation coefficient for any two contexts within the $\ell$-th data type, and $r$ was the parameter controlling the strength of cross-omics/between-data correlations among shared/similar contexts.

After simulating the individual-level data, we obtained all $Z$-scores $\{z_{ij,\ell}\}$'s by performing a simple linear regression for each predictor and its simulated response variable. A similar simulation framework could be used to generate $Z$-scores for three or more data types ($L \geq 3$).

## Cis-e/mQTL input association statistics

We obtained the single-tissue cis-mQTL and cis-eQTL association statistics from GTEx portal[5,23]. Those QTL statistics were obtained using FastQTL[33], adjusting for top five genotypic principal components, biological gender, Sequencing platform (Illumina HiSeq 2000 or HiSeq X), Sequencing protocol (polymerase chain reaction, PCR; PCR-based or PCR-free), and a set of variables generated using the method of probabilistic estimation of expression residuals (PEER)[53]. For cis-mQTL analysis integrating eQTL maps, we included both lead and secondary mQTL variants for each CpG site within a 500KB cis-window size. Each CpG site was assigned to a proximal gene with the nearest TSS[31,32]. We analyzed 204,220 SNP-CpG-gene trios, consisting of 93,681 unique CpG sites and 159,186 unique mQTL variants.

## Trans-e/mQTL input association statistics

In our integrative analysis of trans-e/mQTL associations, we obtained the test statistics for GWAS SNPs associated with at least one out of 80 selected diseases/traits ($P < 5 \times 10^{-8}$). In detail, we selected 80 diseases/traits that have more than 100 risk loci. Those diseases/traits were related to brain function (e.g., Alzheimer's disease), artery tissues (e.g., coronary artery disease), heart function (e.g., atrial fibrillation) or cancers (e.g., prostate cancer). The full list of diseases and traits used was provided in Supplementary Data 1. Similar to cis-eQTL and cis-mQTL analyses, we adjusted for the same set of covariates when performing trans-eQTL and trans-mQTL analyses, respectively. We tested trans-association for SNP-gene pairs from different chromosomes[54,55] and obtained the association statistics for trans-eQTLs and trans-mQTLs in 28 and nine GTEx tissues, respectively. The list of tissues and sample sizes was given in Supplementary Tables 1–2.

## Differential expression analysis of disease-relevant genes in brain layers

In the integrative analysis of spatial transcriptomics data with multi-tissue eQTLs, we used LIBD DLPFC data generated using 10× Visium[45] that contained 12 samples from three adult donors. The original study provided manual annotations for the tissue layers based on the cytoarchitecture. For each of 12 samples, we performed differential expression analysis using beta-Possion GLM model[46] to obtain $Z$-scores of each gene. We compared the expression profiles of spots in a layer with those from the rest of spots in other layers for each gene, and obtained 12 matrices of differential expression statistics with each row being a gene and each column corresponding to a layer. Additionally, we obtained the summary statistics for cis-eQTLs from 13 GTEx brain tissues. For each sample, the two sets of summary statistics were matched through gene names. We conducted a X-ING analysis for each sample. In each analysis, we integrate the spatially differential expression statistics of the sample with 13 sets of GTEx brain eQTL statistics, and obtained the posterior probabilities of having cis-association and spatially differential expression.

More specifically, we analyzed 85,944 GWAS SNPs ($P < 5 \times 10^{-8}$) associated with diseases/traits that have more than 100 risk loci. There were 15,244 cis-genes available in GTEx data for those examined GWAS SNPs[23]. For the 85,944 examined GWAS SNPs, we analyzed a total of 1.6 million SNP-gene pairs matched in 13 GTEx brain tissues and differential expression test statistics for genes among 12 DLPFC samples. In analyzing each of the 12 LIBD samples, we applied X-ING on a 1.6M (SNP-gene pairs) by 7 (layers) matrix of $Z$-scores, and a 1.6M × 13 matrix of $Z$-scores for 13 GTEx brain tissues. Note that 4 of the 12 samples contained only five manually annotated layers (i.e., a 1.6M × 5 matrix of $Z$-scores). At the 90% posterior probability cutoff, we obtained the genes with brain-layer-specific expression levels (≥1 nonzero spatial differential statistic) and also being in cis-association with disease risk loci (≥1 nonzero brain eQTL statistic) in the examined sample (Supplementary Data 2–3). We studied the concerted association and enrichment patterns across 12 samples to minimize the potential confounding effects due to unknown sample heterogeneity.

## Reporting summary

Further information on research design is available in the Nature Portfolio Reporting Summary linked to this article.

## Data availability

The GTEx data (v8) used in this study are available in dbGaP under accession number phs000424.v8.p2. DNAm normalized data is available at GEO (GSE213478). Summary statistics of cis-mQTLs are available at the eGTEx Portal (https://gtexportal.org/home/downloads/egtex/methylation). The GTEx SNPs from GWAS Catalog are available at https://www.ebi.ac.uk/gwas/. The eQTLGen data released by eQTL-Gen Consortium are available at https://www.eqtlgen.org. The GoDMC data released by the Genetics of DNA Methylation Consortium are available at http://www.godmc.org.uk. The FUSION data are available through FUSION Skeletal Muscle Study portal: https://www.ebi.ac.uk/birney-srv/FUSION/. The DLPFC data released by LIBD are available at https://research.libd.org/spatialLIBD/. The CommonMind Consortium data are available via access request to the CommonMind Consortium Knowledge Portal: https://doi.org/10.7303/syn2759792. Data to generate figures is available at https://github.com/ylustat/XING-Analysis/tree/main/Data.

## Code availability

The code for X-ING is available at https://github.com/ylustat/X.ING[56]. The code to reproduce the analysis can be found at https://github.com/ylustat/XING-Analysis/tree/main/Code.

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

## Acknowledgements

We thank the GTEx Consortium. The research of L.S.C. and Y.L. was supported by NIH 2R01GM108711, R35ES028379 and 1R01CA229618. J.L. was supported by National Natural Science Foundation of China No. 12371283, and University Development Fund (UDF01003033) from The Chinese University of Hong Kong-Shenzhen. Y.L. was also supported by Susan G. Komen®TREND21675016. B.L.P. was supported by R35ES028379.

## Author contributions

L.S.C. conceived the project. L.S.C., J.L., and Y.L. developed the methods and wrote the manuscript. Y.L. analyzed the data. M.O. contributed to the data analysis. M.O. and B.L.P. provided valuable suggestions to the development of the methods and analyses. All the authors reviewed the final manuscript.

## Competing interests

The authors declare no competing interests.
