## [Peer Review File · Nature Communications]

Integrative cross-omics and cross-context analysis elucidates molecular links underlying genetic effects on complex traitsReviewer #1 (Remarks to the Author):

See attached pdf file.

Title: Integrative cross-omics and cross-context analysis elucidates molecular links underlying genetic effects on complex traits

Authors: Y Lu, M Oliva, B L Pierce, J Liu and L S Chen

Summary The authors propose X-ING, a novel Bayesian method for identifying statistically significant associations between different omics variables (eg SNPs and cis-genes, SNPs and cis-CpG sites). Instead of averaging across diverse tissue or cell types as in the conventional approach, X-ING preserves the tissue or cell type specific information and borrow information across different types of omics, resulting in a ‘cross-omics’ and ‘cross-context’ analysis. The Bayesian framework imposes sparsity on the posterior results, which helps to identify truly non-zero associations. The probabilistic nature of their method also allows users to make probabilistic statements about the identified non-zero associations (ie post-selection inference).

My major concerns pertain to the somewhat insufficient justification of the statistical methodology by simulation studies. One of the most prominent features of X-ING is to obtain a low-rank approximation of \mathbf{U}_ℓ , the latent matrix that plays a key role in the Bayesian hierarchical model. However, from the existing simulation results, it is not immediately clear what are the benefits of using such a low-rank approximation. To compare the performance of X-ING to that of competing methods, the authors considered 2 simulation settings. However, it seems that more settings are necessary to further identify when and how significant X-ING outperforms competing methods.

Major comments

1. As someone who works on multi-omics integration but does not work on the GTEx data, I find the introduction to the background of the problem a bit difficult to understand. From the abstract, it is not clear what is the difference between ‘cross-omics’ and ‘cross-context’, which makes it difficult for some readers to fully understand the context upon reading the abstract. Therefore, some brief explanation about this is needed in the abstract. More explanations on why integrating summary statistics is preferred to integrating the original data can be given. In the Results section (pp. 3–4), description of the problem can be made more accessible to a wider readership. Upon the first reading, it is not immediately clear how a row of the

$M \times K_\ell$ matrix can be a ‘trio’. (I think Figure 1(a) is trying to explain this but more explicit explanations in the text would be helpful.) More explicit explanations can also be made about the ‘Z-scores’ $\tilde{z}_{i,\ell}$, since ‘Z-scores’ alone is overly general and the reader might need to be reminded what particular Z-scores are being referred to here, and how they could be interpreted as ‘association scores’.

2. How should the number of components p_ℓ and q_ℓ for GCCA and PCA be selected in practice? The sensitivity analysis in S2.2 seems to suggest that the choice of p_ℓ and q_ℓ do not matter. But if this is indeed the case, then it seems to suggest that the low-rank approximations are altogether unnecessary, since reducing the number of components/ranks does not really affect the AUC? If this is not the case, then we still need at least a rule of thumb for selecting (the range of) p_ℓ and q_ℓ ?
3. More comments regarding p_ℓ and q_ℓ . In the results shown in Figs S10 and S11, what is the largest possible number of CCs and PCs? That is, what is the values of the K_ℓ ’s in these studies? This is a key information, since it helps us to understand the benefits of using low-rank approximations. For example, if $K_\ell = 10$, then Figs S10 and S11 seems to suggest that, except for a few cases when $\rho_1 = 0.7$, using low-rank approximations do not help improve the AUC? For the case $\rho_1 = 0.7$, the AUC seems to be higher when we take larger p_ℓ and q_ℓ , which also suggests that we should not use lower rank models.
4. Supp figures S1 and S2 seem to suggest that, when the simulation parameters $\rho_1 = \rho_2 = 0$, different models have very similar ROC curves; for the particular choice $\rho_1 = \rho_2 = 0.7, r = 0.2$, X-ING has the best performance. This seems to suggest that X-ING is not dominantly better than other competing methods for all settings, but only some settings? To conduct a more comprehensive assessment of when X-ING outperforms other methods, more simulation studies with other choices of ρ_1, ρ_2 and r values are needed.

Minor comments

1. On p. 4, the role of $\gamma_{i,\ell}$ can be more explicitly explained. I think the reason for introducing the $\gamma_{i,\ell}$ ’s is that we assume sparsity of truly non-zero effects? (This

seems a standard assumption in the literature, but needs to be made explicit for a wider readership.)

2. Is model (1) already used in previous works? If so, any references?
3. A possible typo on p. 5 (below Eq (2)): it seems that $\mathbf{u}_{0,\ell} \in \mathbb{R}^{K_\ell}$ and $u_{0j,\ell}$ is a scalar?
4. Lines 105–106 on p. 5: what does ‘the latent matrix \mathbf{U}_ℓ consists of two low-rank components, $\mathbf{U}_{\ell L}$ and $\mathbf{U}_{\ell K}$ ’ precisely mean? From the method section, it seems to simply mean that $\mathbf{U}_\ell = \mathbf{U}_{\ell L} + \mathbf{U}_{\ell K}$? Explicitly stating this would probably be better for clarity, since the reader might wonder if ‘low-rank components’ refer to something like $\mathbf{U}_\ell = \mathbf{U}_{\ell L}\mathbf{U}_{\ell K}^\top$, as in low-rank matrix approximation.
5. Different posterior probability cutoffs (80% and 90%) are used on p. 8 and p. 12. What motivated the choices of these different cutoffs for different case studies?
6. What is the definition of $\delta_0(\cdot)$ on p. 15? Dirac delta function?
7. On p. 15 and elsewhere, $\text{Pr}(\cdot)$ is used to denote probability density/likelihood function. However, in the statistics literature, $\text{Pr}(\cdot)$ is usually used to denote ‘the probability of’ instead of the density function. As a standard notation in Bayesian statistics, $p(\cdot)$ can be used here instead of $\text{Pr}(\cdot)$.
8. In the design of simulation studies (pp. 18–19), the covariance matrix $\mathbf{\Omega}$ has a specific group structure, represented by Eqs (8) and (9). What is the motivation for $\mathbf{\Omega}$ to take this structure? Are we mimicking some scenario commonly seen in the real data? Can other structures be considered?
9. Inconsistency of notations (referring to Supp). $\mathbf{U}_{\text{res},\ell}$ on p. 6 in is not defined on p. 5. It seems more reasonable to follow the use of notation as on p. 6, ie define $\mathbf{U}_{\text{res},\ell} = \mathbf{U}_\ell - \mathbf{U}_{\ell L}$, which is not necessarily low-rank, and define $\mathbf{U}_{\ell K}$ as the low-rank approximation of $\mathbf{U}_{\text{res},\ell}$, so that $\mathbf{U}_{\ell K} \approx \mathbf{U}_{\text{res},\ell}$.

Reviewer #2 (Remarks to the Author):

My comments are included in an attached document.

Integrative cross-omics and cross-context analysis elucidates molecular links underlying genetic effects on complex traits

Yihao Lu, Meritxell Oliva, Brandon L. Pierce , Jin Liu, and Lin S. Chen

Paper Review

General Comments

This paper proposed X-ING to learn the sparsity patterns of multiple statistics matrices formed by cis-genetic effects of methylome, transcriptome, and other measurements. The novelty of X-ING in terms of statistical model is assuming the sparsity pattern of each statistics matrix is determined by a low-rank latent matrix, as well as employing canonical correlation analysis (CCA) to leverage the extract shared by these multiple latent matrices. An efficient EM algorithm is derived to implement X-ING in practice, which uses variational inference to reduce the computation burden. Through numeric simulations, the authors demonstrated that X-ING is the sole method that can leverage information from the correlations among statistics matrices. On the one hand, X-ING is able to resolve the cross-omics and cross-context problems arising with the release of eGTE_x data. On the other hand, X-ING introduces a unique insight into genetics, i.e., many high-dimensional problems can be solved by mining their low-dimensional structures, which can be immediately applied to other practical problems.

Main Comments

My main concerns are around the demonstration of X-ING.

1. In the current version, two key assumptions of X-ING, i.e., the sparsity pattern of each statistics matrix is determined by a low-rank latent matrix, as well as CCA is able to trade off the information shared by these multiple latent matrices, do not have statistical models. According to my knowledge, both PCA and CCA can be explained by the factor model., see e.g., Stock and Watson (2002) and Gao et al, (2017). To make X-ING statistically complete, it is better to characterize the statistical models of these two key assumptions.
2. The authors presented various applications of X-ING which revealed novel disease patterns and genetic architectures. In the current version, it is not clear under what outputs of X-ING, the corresponding inferences of X-ING were made. For example, in GWAS, we determine if an SNP has an association with a trait based on the significance level of the corresponding effect size estimate. What are the counterparts in X-ING? An additional paragraph is needed to introduce them.
3. It is worth distinguishing X-ING and its competitors. I guessed that X-ING is the first method to learn the sparsity patterns of multiple statistics matrices which can leverage information from the correlations among statistics matrices. On the other hand, the idea of joint multiple tissues modeling has also been adopted by some methods to improve power and accuracy, e.g., Hu et al. (2019), although these methods focus on other genetic problems. Hence, it is better to compare X-ING with the methods using joint analysis in terms of statistical ideas.

Minor Comments

The followings are the minor comments.

1. VI is an optimization method that use a common distribution q to approximate the true distribution p optimally. It is better to indicate what class of q used to approximate p in supplemental materials.
2. Equation (S4) seems the most important component in the VI-based EM-algorithm. It may be better to clear show the detailed derivation, which can not only prove the validity of the algorithm but also facilitate other researchers to promote it.
3. According to the deduction in supplemental materials, X-ING does not need to estimate $\tilde{z}_{ij,l}$ and $\gamma_{ij,l}$. Hence, it is necessary to indicate the output of Algorithm S1 and under what estimates the corresponding inferences are made.
4. It is better to indicate the numbers of M , L , and K_l in each application, which facilitates the readers to understand what genetic problem X-ING is handling.
5. In the discussion, the authors stated that X-ING cannot efficiently handle the case in which $K_l \geq 50$ and $L \geq 5$. What is the computational burden of X-ING in this case?

Reference

- Gao, C., Ma, Z. and Zhou, H.H., 2017. Sparse CCA: Adaptive estimation and computational barriers. *The Annals of Statistics*, 45(5), pp.2074-2101.
- Hu, Y., Li, M., Lu, Q., Weng, H., Wang, J., Zekavat, S.M., Yu, Z., Li, B., Gu, J., Muchnik, S. and Shi, Y., 2019. A statistical framework for cross-tissue transcriptome-wide association analysis. *Nature genetics*, 51(3), pp.568-576.
- Stock, J.H. and Watson, M.W., 2002. Forecasting using principal components from a large number of predictors. *Journal of the American statistical association*, 97(460), pp.1167-1179.

Reviewer #3 (Remarks to the Author):

Lu et al. propose a multi-tissue, multi-omics data analysis approach, X-ING, to simultaneously detect the sharing of genetic association patterns between tissues and between omics types using (only) summary-level single-SNP association summary statistics. While the approach is well-motivated, and the ideas are intuitive and promising, I find that the writing of the manuscript is inadequate in a few key areas. The biological interpretation of the modeling decisions and computational output is particularly lacking.

The authors claim that the joint modeling of different types of omics data enables detecting "co-occurring" of different kinds of molecular QTLs (line 57, page 3). Does it mean the proposed method performs colocalization analysis if only a single tissue is used? If yes, I think the problem is grossly simplified by ignoring LD between genetic variants in the proposed model. This issue is critical to interpreting the cross-omics patterns output by the method. Given the input and model assumption, it is clearly inappropriate to infer colocalization. The authors need to provide a valid biological interpretation. On the same note, I think the simulation scheme, which only considers independent predictors, is also oversimplified.

Some important areas of the model are not clearly explained. i) How is the correlation matrix, R_I , in Eqn (1) estimated? (Note that there is a claim that this correlation matrix accounts for potential sample overlapping across tissues, line 89, page 4, does the required sample overlapping information is needed in addition to the z-scores?) ii) what does the following statement mean: "The latent matrix U_I consists of two low-rank components. $U_{\{IL\}}$ and $U_{\{IK\}}$ "? Does it imply $U_I = U_{\{IL\}} + U_{\{IK\}}$? Can $U_{\{IL\}}$ and $U_{\{IK\}}$ be uniquely identified? iii) The element of the U_I matrix is formulated in a logistic function in Eqn (2), suggesting that they are interpretable. How do these values connect to the loadings from the proposed PCA and CCA procedure, which are generally considered not directly interpretable?

As the proposed method emphasizes inferring the binary association status, I wonder if the consistency of the effect sizes between tissues is explicitly considered. In a hypothetical case, if an eQTL shows opposite genetic effects between tissues, would they still regard "sharing" genetic effects? What about cross omics-types, where the directions of genetic effects are not directly comparable?

Does the proposed method report the joint posterior distribution of all γ values? If not, how is the sharing between tissues/omics types computed?

It is unclear whether the comparisons with the existing methods are meaningful in simulations. The authors state, "some of those existing methods were proposed for different purposes, and those methods can be adapted to the integrative analysis of summary data from multiple contexts for comparison purposes." To my knowledge, none of the methods is designed for the simulation setting. Worse, they do not even intend to answer the same biological questions. I am unsure what kind of adaptation can put them on an equal footing as the proposed method.

I am also concerned with the strategy of input data construction used in their data application. The analyzed SNPs are selected based on their associations with pre-selected diseases/traits. As such, it is highly problematic to interpret a chosen variant as eQTL or mQTL, even if the corresponding z-scores are highly significant. (They could tag the nearby eQTLs/mQTLs, but themselves have no genetic effects). This issue would undermine all the subsequent claims on mQTLs and eQTLs.

Reviewer #1:

Summary

The authors propose X-ING, a novel Bayesian method for identifying statistically significant associations between different omics variables (eg SNPs and cis-genes, SNPs and cis-CpG sites). Instead of averaging across diverse tissue or cell types as in the conventional approach, X-ING preserves the tissue or cell type specific information and borrow information across different types of omics, resulting in a ‘cross-omics’ and ‘cross-context’ analysis. The Bayesian framework imposes sparsity on the posterior results, which helps to identify truly non-zero associations. The probabilistic nature of their method also allows users to make probabilistic statements about the identified non-zero associations (ie post-selection inference).

My major concerns pertain to the somewhat insufficient justification of the statistical methodology by simulation studies. One of the most prominent features of X-ING is to obtain a low-rank approximation of U_ℓ , the latent matrix that plays a key role in the Bayesian hierarchical model. However, from the existing simulation results, it is not immediately clear what are the benefits of using such a low-rank approximation. To compare the performance of X-ING to that of competing methods, the authors considered 2 simulation settings. However, it seems that more settings are necessary to further identify when and how significant X-ING outperforms competing methods.

Major comments

1. As someone who works on multi-omics integration but does not work on the GTEx data, I find the introduction to the background of the problem a bit difficult to understand. From the abstract, it is not clear what is the difference between ‘cross-omics’ and ‘cross-context’, which makes it difficult for some readers to fully understand the context upon reading the abstract. Therefore, some brief explanation about this is needed in the abstract. More explanations on why integrating summary statistics is preferred to integrating the original data can be given. In the Results section (pp. 3-4), description of the problem can be made more accessible to a wider readership. Upon the first reading, it is not immediately clear how a row of the $M \times K_\ell$ matrix can be a ‘trio’. (I think Figure 1 (a) is trying to explain this but more explicit explanations in the text would be helpful.) More explicit explanations can also be made about the ‘Z-scores’ $\tilde{z}_{i,\ell}$, since ‘Z-scores’ alone is overly general and the reader might need to be reminded what particular Z-scores are being referred to here, and how they could be interpreted as ‘association scores’.

Our response #1:

Those are very good suggestions. Following the reviewer’s suggestions, we rewrote the Abstract, the Introduction, and the Results sections accordingly to emphasize the points raised by the reviewer. Below, we highlight a few key sentences in each section.

1) In the Abstract, we added the following sentence to make the transition before introducing cross-omics and cross-context analysis. “The genetic effects on functionally related ‘omic’ traits often co-occur in relevant cellular contexts (e.g., tissues). The Genotype-Tissue Expression (GTEx) Project studies the effects of variants on gene expression, DNA methylation, and other omic traits across many human tissues.” We also clarified that “X-ING takes as input multiple matrices of association effects/statistics obtained from multi-omics data types, where each matrix consists of columns representing different cellular contexts/tissues/cell types.”

2) In the Introduction, we added the following sentence to explain the rationale of integrating summary statistics: “To efficiently leverage the huge amount of existing data, summary statistics provide a convenient way with secured privacy in sharing data from a wide range of studies.” See

page 2.

3) In the Results section, we added more descriptions of the motivating problem: "Recently, the GTEx consortium released the single-tissue mQTL summary statistics in nine selected tissue types from 424 GTEx participants. Due to the limited tissue-specific sample sizes, the detection of mQTLs is underpowered compared with existing eQTL maps. Moreover, a large majority of mQTLs lack functionality¹. It has been reported that eQTLs often co-occur with mQTLs². The joint analysis of associations of a genetic variant to a cis-gene and a cis-CpG site (a tested trio) could enhance the power and precision in detecting mQTLs, and facilitate the functional interpretations. We develop an integrative association method, X-ING, to jointly analyze multi-tissue mQTL and eQTL statistics, as illustrated in Fig. 1a-c." See pages 3-4, the first paragraph of the Results section.

4) We also added a description of the input 'Z-scores': "The input association summary statistics are Z-statistics obtained from single-tissue e/mQTL analysis. X-ING models the joint association patterns of Z-statistics (as Z-scores) for tested trios across omics data types and tissues. X-ING assumes those Z-scores from K_ℓ tissues, $\mathbf{z}_{i,\ell}$, following a multivariate normal distribution ..." See page 4.

2. How should the number of components p_ℓ and q_ℓ for GCCA and PCA be selected in practice? The sensitivity analysis in S2.2 seems to suggest that the choice of p_ℓ and q_ℓ do not matter. But if this is indeed the case, then it seems to suggest that the low-rank approximations are altogether unnecessary, since reducing the number of components/ranks does not really affect the AUC? If this is not the case, then we still need at least a rule of thumb for selecting (the range of) p_ℓ and q_ℓ ?

Our response #2:

This is a good question. We suggest choosing the number of components p_ℓ and q_ℓ for canonical correlation analysis (CCA) and principal component analysis (PCA) using parallel analysis (PA)^{3,4}. We would like to clarify that the sensitivity analysis in S2.2 aims to show that choosing p_ℓ and q_ℓ within a reasonable range led to similar performance of X-ING. We added additional details to illustrate that when there is no low-rank approximation or choosing p_ℓ and q_ℓ outside the range may lead to worse performance.

Parallel analysis (PA) is widely used for determining the number of principal components (PCs) or factors that should be retained in factor analysis^{3,4}. Sequentially, it generates a random data matrix, performs PCA, and compares the eigenvalues from the real data to the distribution of eigenvalues from the random data. Components are retained if their eigenvalues exceed a certain percentile in the random data eigenvalues, suggesting more variance explained by the retained components than those from expected by chance. PA was also adapted to the context of CCA⁵. Specifically, to select the number of canonical coefficients, p_ℓ , we first performed CCA on two modulation matrices, \mathbf{U}_1^* and \mathbf{U}_2^* . We then generated and analyzed randomly generated data with the same dimension as the original data. Repeating this process 1000 times, we calculated a 99th percentile cutoff of the random squared canonical correlations. Pairs of canonical variates were retained if their canonical correlation exceeded this cutoff. Similarly, to select the number of PCs, q_ℓ , we performed PA on the residual matrices after subtracting the canonical components. When analyzing large data with millions of rows, the computation of the PA analysis may be extensive. One may randomly select a subset of trios (we recommend 50,000) to determine p_ℓ and q_ℓ . As an example, Fig. R1 illustrated how we selected the number of canonical coefficients (CCs) as $p_\ell = 4$ and the number of principal component (PCs) as $q_\ell = 3$ in the cis-e/mQTL analysis of GTEx data.

We also conducted additional simulation studies by varying p_ℓ or q_ℓ . See a detailed description

Figure R1: Scree plots of (a) squared canonical correlations (y-axis) versus the index of canonical correlation coefficients (x-axis) and (b) eigenvalues (y-axis) versus the index of PCs (x-axis) from the cis-e/mQTL analysis of GTEx data.

of the simulation setting in our Reponse # 3 below. As shown in Fig. R2, X-ING achieved similar AUCs for detecting/predicting nonzero QTL effects when choosing p_ℓ or q_ℓ within a certain range. When p_ℓ or q_ℓ is chosen too small (close to 0) or too large (close to full rank), there is no low-rank approximation and the AUCs are much lower.

In summary, we recommend choosing p_ℓ or q_ℓ using parallel analysis. If computation is a concern, PA can be done using randomly selected subsets of trios ($\sim 50,000$) instead of the entire data. As shown in Fig. R2 and the sensitivity analysis in S2.2, the low-rank approximation is necessary to capture major effect-sharing patterns, and X-ING is robust to the choice of p_ℓ or q_ℓ within a reasonable range.

Changes we made: We added the description for choosing p_ℓ or q_ℓ using PA. See page 6. We updated the simulations and expanded the discussion on the choice. We discussed the importance of having the low-rank approximation and stated that X-ING is robust to the choice within a reasonable range. We updated S2.2, Fig. S9 in the Supplementary Materials (see page 11 and 25).

Figure R2: Comparison of AUCs using X-ING with different choices of (a) #CC and (b) #PC. The low-rank approximation (i.e., choosing a small but nonzero number of #CC and #PC) is necessary, and X-ING is robust to the choice of #CC and #PC within a certain range.

3. More comments regarding p_ℓ and q_ℓ . In the results shown in Figs S10 and S11, what is the largest possible number of CCs and PCs? That is, what is the values of the K_ℓ 's in these studies?

This is a key information, since it helps us to understand the benefits of using low-rank approximations. For example, if $K_\ell = 10$, then Figs S10 and S11 seems to suggest that, except for a few cases when $\rho_1 = 0.7$, using low-rank approximations do not help improve the AUC? For the case $\rho_1 = 0.7$, the AUC seems to be higher when we take larger p_ℓ and q_ℓ , which also suggests that we should not use lower rank models.

Our response #3:

In Figs. S10 and S11, the data were generated with a total of 20 tissues/contexts for each data type, $K_\ell = 20$. The results suggested that the performance of X-ING was similar when choosing p_ℓ and q_ℓ within the suggested range of 2 to 8. The largest possible number of CCs was $\min(K_\ell)$ for $\ell = 1, 2$, and here was 20. The largest possible number of PCs was K_ℓ for each data type ℓ .

We further examined the impact of choices of p_ℓ and q_ℓ on the AUC of X-ING. Simulations were conducted with $K_1 = 40$, $K_2 = 40$, and the proportion of variance in the response variable that can be explained by predictors (θ_ℓ) varying from 0.1 to 0.2. The performance of X-ING was assessed for a wide range of #CC in Fig. R2a. The AUCs of X-ING were similar when #CC was within the range of 1-6, and the PA recommended number was $p_\ell = 3$. Subsequently, we assessed the AUCs of X-ING on data 1 using a wide range of #PC in Fig. R2b. X-ING achieved similar AUCs when #PC was in the range of 1-6, and the PA suggested number $q_\ell = 3$.

In Fig. R2, when both p_ℓ and q_ℓ were set to zero, i.e., $\mathbf{U}_\ell = \mathbf{0}$, the model reduced to the starting model with fixed context-specific priors and the AUCs were substantially lower compared with those from the suggested #CC and #PC. When setting $\mathbf{U}_\ell = \mathbf{0}$, the algorithm does not borrow information across data types nor tissue types. When setting p_ℓ or q_ℓ to full ranks with no constraint imposed on \mathbf{U}_ℓ , the model is over-parameterized. The AUCs started to decrease when choosing larger than suggested #CC and #PC, reflecting the impact of over-parameterization. From a regularization perspective, selecting p_ℓ and q_ℓ in their support resembles the regularization on the rank of matrices. Therefore, the low-rank approximation is necessary and captures the major patterns shared across data types and cellular contexts/tissues.

Changes we made: We updated the simulations. We updated S2.2 and Fig. S9 in the Supplementary Materials (see page 11 and 25) to clarify the importance of having the low-rank approximation and stated that X-ING is robust to the choice within a reasonable range.

4. Supp figures S1 and S2 seem to suggest that, when the simulation parameters $\rho_1 = \rho_2 = 0$, different models have very similar ROC curves; for the particular choice $\rho_1 = \rho_2 = 0.7$, $r = 0.2$, X-ING has the best performance. This seems to suggest that X-ING is not dominantly better than other competing methods for all settings, but only some settings? To conduct a more comprehensive assessment of when X-ING outperforms other methods, more simulation studies with other choices of ρ_1, ρ_2 and r values are needed.

Our response #4:

Sorry for the confusion. Fig. S2 showed that all the methods have similar performance when there are no shared patterns ($\rho_1 = \rho_2 = 0$) across contexts/tissues/data and no information to borrow. Different data/contexts are biologically independent. In that setting, all methods performed similarly to single-context analysis.

Fig. S1 showed that the X-ING method performed better compared to other methods when there were shared patterns in the latent matrix \mathbf{U}_ℓ , $\ell = 1, 2$. In addition to Fig. S1, we conducted other simulations to evaluate the performance of X-ING versus competing methods on data with other choices of ρ_1, ρ_2 , and r values. The results are shown in Fig. 2b & 2c of the manuscript.

To further address the reviewer's concern, we added the comparisons of the proposed X-ING ver-

Figure R3: Simulation results comparing X-ING with the multi-tissue eQTL method mash, MT-eQTL and two variations, the starting model X-ING_{starting} without the modeling of cross-context sharing patterns, and the single-omics model X-ING_{single-omics} only modeling context-sharing patterns within a single-omics data type. We considered scenarios with varying strengths of correlation (a) across contexts, ρ , and (b) across data types, r . All compared methods except for X-ING_{starting} can borrow information across context, while only X-ING can borrow information across data types with improved AUC.

Two variations of X-ING, together with the single-omics multi-tissue/context method “mash”, “MT-eQTL” and “Metasoft” (Fig. R3). Specifically, the two variations of X-ING are: 1) the starting model, denoted as X-ING_{starting}, without the modeling of the shared data patterns, and 2) the single-omics model (X-ING_{single-omics}) that models only shared patterns among contexts from the same omics data type. We showed that the starting model without modeling shared patterns, X-ING_{starting}, does not perform well. The performance of the single-omics variation, X-ING_{single-omics}, was similar to that of the existing single-omics multi-tissue method, mash. Mash models multi-tissue QTL effect sizes while X-ING_{single-omics} models the latent binary QTL association status across tissues, both of which allow only cross-context/tissue integration but no borrowing information across data types. When integrating two or more omics data types with increasing strength of data-sharing patterns, r , the proposed X-ING method outperformed single-omics methods (mash and X-ING_{single-omics}).

Changes we made: We rewrote the discussion of results for Figures S1 and S2 to clarify the confusion. We updated the Results section and Figure 2 in the main text (and the Supplementary Materials Fig. S1, S2, S4) to include the comparisons with the two variations of X-ING, X-ING_{starting} and X-ING_{single-omics}. Those comparisons illustrated that X-ING outperformed competing methods by borrowing information across both contexts/tissues and omics data types. X-ING fills a method gap by allowing cross-omics data integration (i.e., the integration of two or more data types).

Minor comments

1. On p. 4, the role of $\gamma_{i,\ell}$ can be more explicitly explained. I think the reason for introducing the $\gamma_{i,\ell}$ ’s is that we assume sparsity of truly non-zero effects? (This seems a standard assumption in the literature, but needs to be made explicit for a wider readership.)

Our response #5:

The role of the latent association status indicator $\gamma_{i,\ell}$ is to model sparsity in true nonzero effects and it allows the joint modeling of nonzero effects across data types even if effect distributions are different. Sparsity of true nonzero effects is a standard assumption in the literature^{6–8} and is important in our model to capture the major data patterns. Accordingly, we have provided a more detailed explanation in the manuscript (see page 5).

2. Is model (1) already used in previous works? If so, any references?

Our response #6:

Yes, similar hierarchical Bayesian models as in (1) have been used in previous works to model Z -scores from multi-tissue eQTL analysis^{6,8}, and Z -scores from GWAS⁷ among others. We added those references in the manuscript and discussed them. See page 5.

3. A possible typo on p. 5 (below Eq(2)): it seems that $u_{0,\ell} \in \mathbb{R}^{K_\ell}$ and $u_{0j,\ell}$ is a scalar?

Our response #7:

Thanks for pointing it out. Yes, it is a typo. It has been corrected.

4. Lines 105-106 on p. 5: what does ‘the latent matrix U_ℓ consists of two low-rank components, $U_{\ell L}$ and $U_{\ell K}$ ’ precisely mean? From the method section, it seems to simply mean that $U_\ell = U_{\ell L} + U_{\ell K}$? Explicitly stating this would probably be better for clarity, since the reader might wonder if ‘low-rank components’ refer to something like $U_\ell = U_{\ell L}U_{\ell K}^\top$, as in low-rank matrix approximation.

Our response #8:

We added the following explanations: “The latent low-rank approximated modulation matrices can be written as $U_\ell = U_{\ell O} + U_{\ell C}, \forall \ell \in \{1, \dots, L\}$, where $U_{\ell O}$ represents the major (i.e., low-rank) omics-shared data structures for data ℓ , and $U_{\ell C}$ represents the major data-specific context-shared structures for data ℓ .”

5. Different posterior probability cutoffs (80% and 90%) are used on p. 8 and p. 12. What motivated the choices of these different cutoffs for different case studies?

Our response #9:

We used a more stringent posterior probability cutoff for differential gene expression analysis (90%) than cis-e/mQTL analysis (80%), while the estimated false discovery rates (FDRs)^{9,10} for the two case studies were similar and both are $< 5\%$ FDR. At the 80% posterior probability threshold, the estimated FDR was 0.076 for the differential gene expression analysis, and the FDR was 0.031 for the cis-e/mQTL analysis. At the 90% posterior probability threshold, the FDR was 0.035 for differential gene expression analysis. Thus, we chose different posterior probability cutoffs for the two studies while keeping the estimated FDR both below 5%. We now included the estimated FDR in the main text (see page 10, 14) to justify the choice of the posterior probability cutoffs.

6. What is the definition of $\delta_0(\cdot)$ on p. 15? Dirac delta function?

Our response #10:

Yes, $\delta_0(\cdot)$ refers to Dirac delta function¹¹. The function is a generalized distribution with

$$\delta_0(x) = \begin{cases} 0, & \text{if } x \neq 0 \\ \infty, & \text{if } x = 0 \end{cases}, \text{ with } \int_{-\infty}^{+\infty} \delta_0(x) dx = 1.$$

We added the description in the main text (see page 17).

7. On p. 15 and elsewhere, $\Pr(\cdot)$ is used to denote probability density/likelihood function. However, in the statistics literature, $\Pr(\cdot)$ is usually used to denote ‘the probability of’ instead of the

density function. As a standard notation in Bayesian statistics, $p(\cdot)$ can be used here instead of $\text{Pr}(\cdot)$.

Our response #11:

We made the suggested changes in the manuscript and the Supplementary Materials.

8. In the design of simulation studies (pp. 18-19), the covariance matrix Ω has a specific group structure, represented by Eqs (8) and (9). What is the motivation for Ω to take this structure? Are we mimicking some scenario commonly seen in the real data? Can other structures be considered?

Our response #12:

The structure is motivated by the real structures of GTEx data (see Fig. S41 of the GTEx paper¹²). Biologically related tissue groups were identified, and sharing of effects among related tissues was reported. Yes, other structures and effect-sharing patterns can be captured by the X-ING model.

9. Inconsistency of notations (referring to Supp). $U_{\text{res},\ell}$ on p. 6 in is not defined on p. 5. It seems more reasonable to follow the use of notation as on p. 6, ie define $U_{\text{res},\ell} = U_{\ell} - U_{\ell L}$, which is not necessarily low-rank, and define $U_{\ell K}$ as the low-rank approximation of $U_{\text{res},\ell}$, so that $U_{\ell K} \approx U_{\text{res},\ell}$

Our response #13:

The reviewer is correct. We made the corresponding changes in the supplementary materials to make the notations consistent.

Reviewer #2:

General Comments

This paper proposed X-ING to learn the sparsity patterns of multiple statistics matrices formed by cis-genetic effects of methylome, transcriptome, and other measurements. The novelty of X-ING in terms of statistical model is assuming the sparsity pattern of each statistics matrix is determined by a low-rank latent matrix, as well as employing canonical correlation analysis (CCA) to leverage the extract shared by these multiple latent matrices. An efficient EM algorithm is derived to implement X-ING in practice, which uses variational inference to reduce the computation burden. Through numeric simulations, the authors demonstrated that X-ING is the sole method that can leverage information from the correlations among statistics matrices. On the one hand, X-ING is able to resolve the cross-omics and cross-context problems arising with the release of eGTEx data. On the other hand, X-ING introduces a unique insight into genetics, i.e., many high-dimensional problems can be solved by mining their low-dimensional structures, which can be immediately applied to other practical problems.

Main Comments

My main concerns are around the demonstration of X-ING.

1. In the current version, two key assumptions of X-ING, i.e., the sparsity pattern of each statistics matrix is determined by a low-rank latent matrix, as well as CCA is able to trade off the information shared by these multiple latent matrices, do not have statistical models. According to my knowledge, both PCA and CCA can be explained by the factor model., see e.g., Stock and Watson (2002) and Gao et al, (2017). To make X-ING statistically complete, it is better to characterize the statistical models of these two key assumptions.

Our response #14:

In the general form of X-ING, we modulate the probability of the latent binary association status, $\gamma_{ij,\ell}$, and link it with the latent modulation matrix capturing major/desirable patterns in the data:

$$\log \frac{p(\gamma_{ij,\ell} = 1 | \mathbf{U}_\ell, \mathbf{u}_{0,\ell})}{p(\gamma_{ij,\ell} = 0 | \mathbf{U}_\ell, \mathbf{u}_{0,\ell})} = U_{ij,\ell} + u_{0j,\ell}, \quad (\text{R1})$$

where $u_{0j,\ell}$ is the context-specific intercept, controlling the sparsity of nonzero effects in the j -th context of the ℓ -th omics data, and $U_{ij,\ell}$ is the element in the latent modulation matrix \mathbf{U}_ℓ for the i -th trio in the j -th context of ℓ -th omics data.

In the motivating e/mQTL analysis, QTL effects are sparse in the genome. We assume a low-rank structure of the latent matrix. The latent matrix can be written as $\mathbf{U}_\ell = \mathbf{U}_{\ell O} + \mathbf{U}_{\ell C}, \forall \ell \in \{1, \dots, L\}$, where $\mathbf{U}_{\ell O}$ represents the major (i.e., low-rank) omics-shared data structures for data ℓ , and $\mathbf{U}_{\ell C}$ represents the major data-specific context-shared structures for data ℓ . We propose to apply CCA on the latent logit-transformed probability matrices \mathbf{U}_ℓ^* to obtain $\mathbf{U}_{\ell O}$, and then apply PCA on the residual matrices after subtracting $\mathbf{U}_{\ell O}$ from \mathbf{U}_ℓ^* to obtain $\mathbf{U}_{\ell C}$.

It should be noted that the latent matrix \mathbf{U}_ℓ may take other forms. For example, we derived two variations of X-ING in the simulations, when $\mathbf{U}_\ell = \mathbf{0}$, and when \mathbf{U}_ℓ is of full rank. Similarly, the multi-tissue QTL method mash⁸ allows data-driven matrices that are estimated from the data as major patterns or user-provided matrices that have simple interpretations to model the structures in the data.

The reviewer raised a good point. Both PCA and CCA can be explained by factor models. Similar to Lock et al.¹³, we could have the omics-shared low-rank matrix rewritten as $\mathbf{U}_{\ell O} = \mathbf{V}\mathbf{W}_{\ell O}$, where $\mathbf{W}_{\ell O} \in \mathbb{R}^{p_\ell \times K_\ell}$ and $\mathbf{V} \in \mathbb{R}^{M \times p_\ell}$ are the loading matrix and the corresponding factors, respectively,

and p_ℓ is the rank of $\mathbf{U}_{\ell C}$. We could also rewrite the data-specific low-rank matrix with rank q_ℓ as $\mathbf{U}_{\ell C} = \mathbf{V}_{\ell C} \mathbf{W}_{\ell C}$, where $\mathbf{W}_{\ell C} \in \mathbb{R}^{q_\ell \times K_\ell}$ and $\mathbf{V}_{\ell C} \in \mathbb{R}^{M \times q_\ell}$ are the loading matrix and corresponding factors, respectively. The latent low-rank modulation matrices in X-ING may take the form:

$$\mathbf{U}_\ell = \mathbf{V} \mathbf{W}_{\ell O} + \mathbf{V}_{\ell C} \mathbf{W}_{\ell C}, \forall \ell \in \{1, \dots, L\}.$$

One may regularize the ranks of the omics-shared and data-specific low-rank matrices, $\mathbf{U}_{\ell O}$ and $\mathbf{U}_{\ell C}$, to estimate the low-rank matrices and obtain an estimate for \mathbf{U}_ℓ .

In the proposed estimation algorithm, we choose to separately estimate $\mathbf{U}_{\ell O}$ and $\mathbf{U}_{\ell C}$ using CCA followed by PCA on the residual matrices for computational concerns and for the interpretability of the obtained CCs and PCs. See Response #23 and Figure R6 for detailed biological interpretations of the obtained CCs and PCs.

Changes we made: Following the reviewer's suggestion, we expanded the discussion of the model in the main text and the choice of the modulation matrix, \mathbf{U}_ℓ . We also explained the rationale of the sparsity assumption on the \mathbf{U}_ℓ matrices for QTL analysis, and the proposed algorithm for separately estimating the omics-shared and data-specific low-rank matrices. See page 6. Note that we changed the notations, with $\mathbf{U}_{\ell O}$ denoting the omics-shared major structures for data ℓ and $\mathbf{U}_{\ell C}$ for data-specific context-shared major structures.

2. The authors presented various applications of X-ING which revealed novel disease patterns and genetic architectures. In the current version, it is not clear under what outputs of X-ING, the corresponding inferences of X-ING were made. For example, in GWAS, we determine if an SNP has an association with a trait based on the significance level of the corresponding effect size estimate. What are the counterparts in X-ING? An additional paragraph is needed to introduce them.

Our response #15:

This is a good question. X-ING is a general integrative association analysis method. It takes as input multiple matrices of association statistics from multiple omics studies each with multiple contexts. The outputs of X-ING are 1) (refined) multi-context posterior means of input statistics, e.g., multi-tissue mQTL effects/statistics, and 2) estimated posterior probabilities of nonzero effects, as illustrated in Fig. 1d in the main text. The posterior probabilities allow flexible inference. For example, in the cis-e/mQTL analysis, we detected mQTLs with multi-tissue effects (PP>0.8 in two or more DNAm tissues). We detected mQTLs with co-occurring associations to cis-expression (with PP>0.8 in DNAm tissues and at least one expression tissue) at the 80% posterior probability cutoffs. One may also calculate FDRs based on the posterior probabilities⁹.

Changes we made: Following the reviewer's suggestion, we included an additional paragraph in the Results section to discuss the outputs and how we make inferences using the outputs of X-ING (see page 4 in the manuscript and Fig. 1d).

3. It is worth distinguishing X-ING and its competitors. I guessed that X-ING is the first method to learn the sparsity patterns of multiple statistics matrices which can leverage information from the correlations among statistics matrices. On the other hand, the idea of joint multiple tissues modeling has also been adopted by some methods to improve power and accuracy, e.g., Hu et al. (2019), although these methods focus on other genetic problems. Hence, it is better to compare X-ING with the methods using joint analysis in terms of statistical ideas.

Our response #16:

The reviewer is correct that X-ING fills a method gap allowing the integrative association analy-

sis of multiple statistic matrices from multiple omics data types each with multivariate contexts (columns).

Changes we made: Following the reviewer's suggestion, we discussed the similarity between the proposed X-ING method and other joint multi-tissue modeling methods, including the multi-tissue QTL method mash⁸, and the unified test for molecular signatures (UTMOST)¹⁴. UTMOST is a Transcriptome-wide association studies (TWASs) method that identifies gene-trait associations by jointly modeling multi-tissue gene expression. Mash, UTMOST and X-ING jointly model multi-tissue/context statistics and leverage shared patterns to increase statistical power compared with single-context analyses. See page 3 in the main text.

Minor Comments

The followings are the minor comments.

1. VI is an optimization method that use a common distribution q to approximate the true distribution p optimally. It is better to indicate what class of q used to approximate p in supplemental materials.

Our response #17:

We used a mean-field variational family. Random variables of different contexts or tested units are assumed to be independent. We revised the Supplementary Materials (see page 3).

2. Equation (S4) seems the most important component in the VI-based EM-algorithm. It may be better to clear show the detailed derivation, which can not only prove the validity of the algorithm but also facilitate other researchers to promote it.

Our response #18:

We added it in the Supplementary Materials (see section S1.1).

3. According to the deduction in supplemental materials, X-ING does not need to estimate $\tilde{z}_{ij,l}$ and $\gamma_{ij,l}$. Hence, it is necessary to indicate the output of Algorithm S1 and under what estimates the corresponding inferences are made.

Our response #19:

We made the suggested changes to Algorithm S1. We expand the discussion of X-ING output and flexible inference based on the estimated posterior probabilities. See response #15, and page 4 in the main text.

4. It is better to indicate the numbers of M , L , and K_l in each application, which facilitates the readers to understand what genetic problem X-ING is handling.

Our response #20:

This is a good suggestion. We made the corresponding changes and included the following numbers in each application.

1. Multi-tissue cis-e/mQTL integrative analysis: We applied X-ING to the cis-eQTL and mQTL association statistics generated on 28 and 9 tissues ($K_1 = 28$, $K_2 = 9$, $L = 2$), respectively. we analyzed 204,220 unique SNP-Gene-CpG trios ($M = 204,220$).

2. Trans-association enrichment analysis: X-ING integrated summary statistics of multi-tissue

trans-eQTLs and trans-mQTLs for each of the 80 diseases/traits ($L = 2$). We incorporated trans-eQTL association statistics from 28 GTEx tissues ($K_1 = 28$) and trans-mQTL statistics from nine GTEx tissues ($K_2 = 9$). The SNP-Gene-CpG trios (M) varied from ~ 3 million (Alzheimer’s disease) to ~ 68 million (type 2 diabetes).

3. Integrative analysis of spatial transcriptomic data and multi-tissue eQTLs: We combined test statistics from spatial differential expression analysis across six brain layers and white matter ($K_1 = 7$) with eQTL statistics from 13 GTEx brain tissues ($K_2 = 13$). We jointly analyzed over 1.6M matched SNP-gene pairs from LIBD and GTEx data ($L = 2, M \sim 1.6$ million).

5. In the discussion, the authors stated that X-ING cannot efficiently handle the case in which $K_l \geq 50$ and $L \geq 5$. What is the computational burden of X-ING in this case?

Our response #21:

When L or K is large, the computation is mainly constrained by the generalized CCA (GCCA) algorithm for a large number of trios. Based on previous literature^{15,16}, when $L = 2$, time complexity of CCA is $O(Md^2)$ where $d = \max(K_1, K_2)$, M is the number of rows for the input matrices. The complexity of PCA is $O(MK_\ell^2)$. When $L > 2$, we performed an empirical runtime analysis with for larger L and K with different numbers of trios (Fig. R4). We used the `rgcca` function in RGCCA Package on the simulated data. The computation time increases quadratically with the number of context, K . When $L \geq 5, K_l \geq 50$ and the number of trios is up to millions, the computation burden is very heavy. We expanded the discussion on computation time (see page 8). On the other hand, Supplementary Fig. S6 and S7 showed the comparison of the computation time of X-ING versus other methods. X-ING is relatively efficient compared with other methods.

Figure R4: The mean of computation time of GCCA on data with different numbers of omics data types (L), contexts (K_ℓ), and trios.

Reviewer #3:

Lu et al. propose a multi-tissue, multi-omics data analysis approach, X-ING, to simultaneously detect the sharing of genetic association patterns between tissues and between omics types using (only) summary-level single-SNP association summary statistics. While the approach is well-motivated, and the ideas are intuitive and promising, I find that the writing of the manuscript is inadequate in a few key areas. The biological interpretation of the modeling decisions and computational output is particularly lacking.

1. The authors claim that the joint modeling of different types of omics data enables detecting "co-occurring" of different kinds of molecular QTLs (line 57, page 3). Does it mean the proposed method performs colocalization analysis if only a single tissue is used? If yes, I think the problem is grossly simplified by ignoring LD between genetic variants in the proposed model. This issue is critical to interpreting the cross-omics patterns output by the method. Given the input and model assumption, it is clearly inappropriate to infer colocalization. The authors need to provide a valid biological interpretation. On the same note, I think the simulation scheme, which only considers independent predictors, is also oversimplified.

Our response #22:

The reviewer raised a good point. We would like to clarify that our proposed method X-ING is an integrative association method. X-ING aims to identify joint associations and the results should not be interpreted as causation. We realized that there has been confusion regarding the wording "co-occurring" in the previous discussions. The co-occurrence of e/mQTLs may or may not involve a common causal variant underlying both QTLs. We made changes throughout the manuscript to make the descriptions more precise. Instead of co-occurring QTLs, we rephrase them as "co-occurring associations" or joint associations when applicable.

Different than colocalization analysis aiming to identify shared causal genetic variants for multiple (molecular or complex) traits, X-ING examines only joint associations of genetic variants to multiple traits (from different omics data and contexts/tissues). X-ING was motivated by the multi-tissue mQTL analysis of GTEx data when integrating with existing eQTL maps. The two main challenges are: 1) mQTL sample sizes are limited and mQTL detections are under-powered, and 2) the vast majority of mQTLs lack functionality. As an integrative association method, X-ING jointly analyzes multi-tissue mQTL association effects with multi-tissue association effects to a cis-gene expression. The rationale is that many functional genetic variants exert their effects on traits/diseases through gene regulation. The cascading effects of genetic variants from the epigenome to the transcriptome and eventually to complex traits and diseases are frequently reported¹⁷. If an mQTL is also associated with the expression levels of a cis-gene, X-ING could leverage the co-occurrence of mQTL effects and its association with the cis-gene to boost confidence in detecting mQTLs in relevant tissues and improve the functional and mechanistic interpretations of the mQTL. But the mQTL may not be the causal eQTL, and could be only in LD with the eQTL.

We agree with the reviewer that inferring colocalization from X-ING's output is inappropriate. That is not our intention either. We made clarifications throughout the manuscript to avoid confusions.

Regarding the simulation scheme, we performed additional simulations to examine the impact of SNP correlations on AUCs for detecting nonzero effects. We simulated correlated SNPs with a block-diagonal LD matrix. For each gene, we simulated 10 cis-SNPs for each block, with a total of 50 blocks. Within the same block, the pairwise correlation among SNPs varied from 0 to 0.4. This simulation mimics a real data analysis with input statistics being effects for candidate QTLs. The estimation of X-ING model assumes the examined SNPs being independent. The simulation

results show that weak correlation among SNPs did not substantially hurt the performance of X-ING (Fig. R5). In practice, we recommend conducting LD pruning on the input data before applying X-ING. An r^2 threshold of 0.1 is recommended.

Changes we made: We made clarification throughout the manuscript that X-ING infers joint associations instead of colocalization. We added the following sentence in the discussion: "X-ING does not perform colocalization analysis." We changed the wording of co-occurring QTLs to co-occurring associations or joint associations. We included Fig. R5 as Supplementary Fig. S8 and discussed the impact of weak LD on the performance of X-ING.

Figure R5: Comparison of AUCs using X-ING for detecting nonzero effects. Simulated data were generated with varying levels of pairwise correlation for SNPs within the same block. Here θ_ℓ represents the variance in expression that can be explained by the SNPs.

2. Some important areas of the model are not clearly explained. i) How is the correlation matrix, \mathbf{R}_ℓ , in Eqn (1) estimated? (Note that there is a claim that this correlation matrix accounts for potential sample overlapping across tissues, line 89, page 4, does the required sample overlapping information is needed in addition to the z-scores?) ii) what does the following statement mean: "The latent matrix \mathbf{U}_ℓ consists of two low-rank components. $\mathbf{U}_{\ell L}$ and $\mathbf{U}_{\ell K}$ "? Does it imply $\mathbf{U}_\ell = \mathbf{U}_{\ell L} + \mathbf{U}_{\ell K}$? Can $\mathbf{U}_{\ell L}$ and $\mathbf{U}_{\ell K}$ be uniquely identified? iii) The element of the \mathbf{U}_ℓ matrix is formulated in a logistic function in Eqn (2), suggesting that they are interpretable. How do these values connect to the loadings from the proposed PCA and CCA procedure, which are generally considered not directly interpretable?

Our response #23:

(i) The correlation matrix, \mathbf{R}_ℓ in Eqn. (1) represents tissue-tissue correlation due to sample overlapping. Existing literature often estimates it using a subset of the input Z-statistics that are likely to be from the null distributions (for example, using only SNPs with $|Z| < 5$ in all tissues to estimate the tissue-tissue correlation matrix)^{8,18}. This correlation matrix due to sample overlapping (under the null, not of interest) is estimated a priori and is taken as known. It is separated from the biological co-occurrence of associations among related tissues (tissue-tissue correlation under the alternative, captured by low-rank factors of logit-transformed matrices in the X-ING model).

(ii) Sorry for the confusion. Yes, $\mathbf{U}_\ell = \mathbf{U}_{\ell O} + \mathbf{U}_{\ell C}$. In terms of the identifiability of the low-rank components $\mathbf{U}_{\ell O}$ and $\mathbf{U}_{\ell C}$, we need to clarify that $\mathbf{U}_{\ell O}$ and $\mathbf{U}_{\ell C}$ are uniquely identified only when the row-space orthogonality between $\mathbf{U}_{\ell O}$ and $\mathbf{U}_{\ell C}$ is met^{13,19}, i.e., when the omics-shared joint structures are unrelated to the data-specific context-shared structures. In our applications, the two matrices $\mathbf{U}_{\ell O}$ and $\mathbf{U}_{\ell C}$ may not be uniquely identifiable. The idea is to capture the desired data patterns to boost the power of detecting the desired types of associations. Those patterns do not need to be orthogonal to each other, as long as their sum is a good approximation and

valid estimate for the logit-transformed latent probability matrix. We link the estimated \mathbf{U}_ℓ matrix with the probabilities for $\gamma_{ij,\ell}$ being nonzero via a logit link (see Eqn. (2)). In other words, the modeling of omics-shared and data-specific patterns may not be uniquely identifiable (and they do not need to be), as long as their sum \mathbf{U}_ℓ is a good approximation of the logit-transformed probabilities of latent indicators and the sum captures the desired patterns.

Figure R6: Absolute values of correlations between the estimated sample-averaged cell-type fractions for the listed cell types and its most correlated PC. The significance of the correlations is labeled. Cell types that show significant correlations with at least one PC in both eQTL and mQTL data are in purple. The sample-averaged cell-type fractions are derived from (a) expression data using CIBERSORTx and (b) DNA methylation data using EpiDISH.

(iii) Our analysis suggests that eigenvectors of covariance matrix of modulation matrices capture biologically meaningful features. In the context of multi-tissue eQTL (28 tissues) and mQTL (9 tissues) analysis, we estimated the sample-averaged cell type fractions using CIBERSORTx²⁰ and EpiDISH²¹ from expression and DNA methylation data, respectively. We then calculated the absolute correlations between the eigenvectors from the modulation matrices of X-ING ($\mathbf{U}_{\ell C}$'s for eQTL and mQTL data) and sample-averaged cell type fractions estimated from individual-level data¹ across tissues. We showed that the eigenvectors are highly correlated with multiple major cell types (Fig. R6). In other words, the major patterns/eigenvectors captured by PCA (similarly for CCA) can be interpreted as the surrogate variables for tissue-tissue dependence due to similar cell type compositions. Similar conclusions have been reported by GTEx and other QTL consortia. In GTEx, expression data derived PEER factors (similar to PCs) are highly correlated with estimated major cell types enrichment scores^{12,22}.

Changes we made: We added the clarification details of the above points in the manuscript (see page 17, 6 and 9) We also included Fig. R6 in Supplementary Figure S10. Note that we changed the notations, with $\mathbf{U}_{\ell O}$ denoting the major omics-shared data structures for data ℓ and $\mathbf{U}_{\ell C}$ representing the major data-specific context-shared structures.

3. As the proposed method emphasizes inferring the binary association status, I wonder if the consistency of the effect sizes between tissues is explicitly considered. In a hypothetical case, if an eQTL shows opposite genetic effects between tissues, would they still regard "sharing" genetic effects? What about cross omics-types, where the directions of genetic effects are not directly comparable?

Our response #24:

This is a good question. Regarding the biological interpretation of opposite eQTL effects, previous literature reported ~7.4% eQTLs have opposite signs in different tissues and even between closely related tissues²³. Those eQTLs showed locational enrichment at the transcription start site and also possible involvement of epigenetic regulation. The biological importance of the opposite eQTL effects was assessed, demonstrating a high proportion (26.9%) of those eQTLs are in LD with GWAS SNPs. The opposite eQTL effects were considered as at least partially due to tissue-specific gene regulation.

In the cross omics-type, not only the directions of genetic effects but also the effect sizes are not directly comparable. X-ING allows the integration of those non-comparable effects by modeling the latent binary status. That is, the presence of true nonzero association effects (even if opposite signs, or different effect sizes) in other data or tissues/contextes would enhance the confidence in detecting true nonzero effects in the current data/context of interest.

The reviewer raised a valid point. If an eQTL shows opposite nonzero genetic effects between tissues, we should not consider them as "sharing genetic effects". Rather, we could still consider them as sharing the presence of nonzero effects, or with co-occurring associations. We changed the wording throughout the manuscript to be more precise.

4. Does the proposed method report the joint posterior distribution of all γ values? If not, how is the sharing between tissues/omics types computed?

Our response #25:

Yes, the posterior probabilities of $\gamma_{ij,\ell} = 1$ for all trios i , in context j , from the ℓ -th omics data are reported as output. Those probabilities can be further thresholded to obtain QTLs with nonzero effects in multiple data/contextes or data/contextes of interest. Fig. 1 in the main text and Supplementary Algorithm 1 illustrates the output. The context-shared and omics-data-shared major patterns of latent binary association status were accounted for during the model fitting process using PCA and CCA (Eqn. (2) in the main text).

5. It is unclear whether the comparisons with the existing methods are meaningful in simulations. The authors state, "some of those existing methods were proposed for different purposes, and those methods can be adapted to the integrative analysis of summary data from multiple contexts for comparison purposes." To my knowledge, none of the methods is designed for the simulation setting. Worse, they do not even intend to answer the same biological questions. I am unsure what kind of adaptation can put them on an equal footing as the proposed method.

Our response #26:

We hear the reviewer's concern about lacking the proper comparison methods. We argue that X-ING allows cross-omics data integration (i.e., two or more summary statistic matrices as input) and it fills a method gap. The most comparable existing method is mash⁸, which is proposed for multi-tissue eQTL analysis (i.e., one summary statistic matrix as input). It jointly analyzes QTL statistics from multiple tissues/contextes of only one omics data type at a time.

Other comparison methods are: MT-eQTL⁶ uses a hierarchical Bayesian model to simultaneously model association statistics from various tissues. Metasoft²⁴ applies a random-effects model to capture effect variation across contexts. Those methods share the similarities of leveraging multi-context data and accounting for major patterns to improve power and estimation. On the other hand, those methods do not allow the integration of effect sizes from different natures/distributions.

To address the reviewer's concern, we included two variations of X-ING as additional comparison methods to illustrate the gains of the cross-omics integration and the model of major shared patterns. The two variations of X-ING are: 1) the starting model, denoted as X-ING_{starting}, without the modeling of the shared data patterns (also no low-rank approximation), and 2) the single-omics model (X-ING_{single-omics}) modeling only shared patterns among contexts from the same omics data type. See Response #4 and Fig. R3 for simulation results. We showed that the starting model without modeling shared patterns, X-ING_{starting}, does not perform well. The performance of the single-omics variation, X-ING_{single-omics}, was similar to that of the existing method single-omics data mash. Mash and X-ING_{single-omics} both allow only cross-context/tissue integration but no borrowing information across data types. The former models effect sizes and the latter models the latent association status, and their performances are similar. With the increasing strength of data-sharing patterns, the proposed cross-omics and cross-context X-ING method outperformed single-omics methods with the integration of two or more omics data types.

Regarding PAINTOR, this method was primarily developed for multi-trait and fine mapping in GWAS studies and may not be directly applicable to our setting. Therefore, we have removed PAINTOR from our comparison.

Changes we made: We revised the simulation section and made the above changes. See page 6 and Figure 2.

6. I am also concerned with the strategy of input data construction used in their data application. The analyzed SNPs are selected based on their associations with pre-selected diseases/traits. As such, it is highly problematic to interpret a chosen variant as eQTL or mQTL, even if the corresponding z-scores are highly significant. (They could tag the nearby eQTLs/mQTLs, but themselves have no genetic effects). This issue would undermine all the subsequent claims on mQTLs and eQTLs.

Our response #27:

This is a valid concern. We appreciate the suggestion and revise the data analysis for cis-e/mQTL analysis. Instead of focusing on disease/trait-associated SNPs, in the revised application we propose to detect multi-tissue cis-mQTLs with the integration of eQTL maps to improve the precision/replication and functionality of detected mQTLs.

Specifically, we obtained the sets of mQTLs from single-tissue mQTL analysis of eGTE¹ including both lead and secondary mQTL variants for each CpG site. We included 93,681 CpG sites and 159,186 unique mQTL variants, forming 204,220 unique SNP-CpG pairs. Each CpG site was assigned to a proximal gene with the nearest transcription start site^{25,26}, forming 204,220 SNP-CpG-gene trios. We applied X-ING to the cis-mQTL association statistics from 9 DNAm tissues and integrated with the cis-eQTL association statistics from 28 expression tissues.

At the 80% posterior probability cutoff, among the 204,220 analyzed SNP-CpG pairs, we identified a total of 143,801 pairs with nonzero mQTL effects in at least two tissues. Among those 143,801 SNP-CpG pairs (corresponding to 112,162 unique mQTL variants), 79,454 (58,158 unique variants) also exhibited nonzero association effect to its cis-gene expression in at least one tissue. In other words, more than half of the reported mQTLs also have nonzero associations to its cis-genes. It should be noted that those mQTLs are not necessarily the causal eQTL variants and could be in LD with the causal eQTLs and have co-occurring association effects.

We further evaluated the replication rates of SNP-CpG pairs with nonzero effects identified by X-ING. In total, 119,401 out of 204,220 analyzed lead/secondary SNP-CpG pairs were also included in the replication data from FUSION (Finland-United States Investigation of NIDDM Genetics)

skeletal muscle study²⁷. Among those 119,401 SNP-CpG pairs, 84,255 have nonzero effects in at least two tissues (PP>0.8) in GTEx data. At the P -value threshold of 6×10^{-7} (with Bonferroni correction)²⁸⁻³⁰, 45.04% (53,780 out of 119,401) of the input SNP-CpG pairs were replicated in the FUSION data (without applying X-ING). In contrast, 55.79% (47,010 out of 84,255) of the SNP-CpG pairs identified by X-ING with multi-tissue effects (in two or more tissues) were replicated in FUSION. Moreover, we further categorized the examined mQTLs as 1) single-tissue mQTLs only, 2) single-tissue mQTLs with co-occurring expression associations, 3) multi-tissue mQTLs only, and 4) multi-tissue mQTLs with co-occurring expression associations. Table R1 shows that the replication rates by tissue type for those four types of mQTLs. Not surprisingly, multi-tissue mQTLs are more likely to be replicated. It is worth noting that mQTLs with co-occurring expression associations have a much higher replication rates than those without. Similar replication results were observed in GoDMC data (Table R2). The replication results demonstrate that by integrating multi-omics association studies and borrowing information across data types, X-ING improves the detection, replication, and functional interpretation of mQTLs.

In the trans-association analysis, we still focus on disease/trait-associated GWAS SNPs. Following the reviewer's suggestion, we revised the interpretation of the results as the GWAS SNPs being in trans-associations to a CpG site and gene expression in a distal region. We studied the patterns of trans-associations instead of reporting individual trans-QTLs. The eQTLGen consortium³¹ also focused on trait-associated SNPs when studying trans-eQTL associations.

Tissue	Sample Size	Total # identified SNP-CpG pairs (PP> 0.8)	Type of effects			
			Single-tissue, no expression association	Single-tissue, with expression association	Multi-tissue, no expression association	Multi-tissue, with expression association
Breast Mammary Tissue	49	13,870	0.08 (18/233)	0.38 (18/47)	0.74 (3,259/4,399)	0.83 (7,619/9,191)
Colon Transverse	189	83,803	0.17 (1,013/6,047)	0.26 (955/3,710)	0.53 (16,532/31,140)	0.63 (26,949/42,906)
Kidney Cortex	47	13,690	0.39 (11/28)	0.40 (6/15)	0.64 (3,489/5,445)	0.76 (6,251/8,202)
Lung	190	74,282	0.09 (427/4,819)	0.19 (464/2,506)	0.54 (14,957/27,733)	0.63 (24,746/39,224)
Muscle Skeletal	42	11,313	0.44 (24/54)	0.75 (24/32)	0.82 (3,193/3,881)	0.91 (6,672/7,346)
Ovary	140	61,840	0.14 (922/6,431)	0.24 (1,067/4,047)	0.30 (11,658/21,215)	0.40 (19,660/30,147)
Prostate	105	51,125	0.12 (258/1,900)	0.27 (273/906)	0.32 (11,386/19,172)	0.42 (20,081/29,147)
Testis	47	9,631	0.11 (47/501)	0.16 (28/179)	0.33 (2,182/3,272)	0.41 (4,102/5,679)
Whole Blood	47	29,719	0.08 (34/442)	0.30 (33/114)	0.34 (6,327/10,869)	0.48 (12,638/18,294)

Table R1: Replication rates in the FUSION data for SNP-CpG associations identified by X-ING (PP>0.8). Proportions and numbers of SNP-CpG pairs identified in GTEx that are also significant in FUSION ($P < 6 \times 10^{-7}$) are shown. Those SNP-CpG pairs were divided into four groups based on the number of tissues and the presence of associations with cis-genes.

Changes we made: Following the reviewer's suggestion, We rewrote the entire analyses section to reflect those changes. We updated the replication studies. We changed the sub-section title to "A multi-tissue cis-mQTL analysis integrating eQTL maps". We changed "eQTLs" to "expression associations" (if appropriate) to make it more precise. Similarly, we changed trans-QTLs to trans-associations when applicable. We made the interpretation clearer and more precise throughout the manuscript. Tables R1 and R2 were included as Table 2 in the main text. We appreciate the suggestion.

Tissue	Sample Size	Total # identified SNP-CpG pairs (PP> 0.8)	Type of effects			
			Single-tissue, no expression association	Single-tissue, with expression association	Multi-tissue, no expression association	Multi-tissue, with expression association
Breast Mammary Tissue	49	24,583	0.07 (31/431)	0.26 (20/78)	0.31 (2,474/8,026)	0.45 (7,259/16,048)
Colon Transverse	189	143,566	0.18 (1,993/10,792)	0.31 (1,836/5,960)	0.31 (17,060/55,019)	0.41 (29,614/71,795)
Kidney Cortex	47	24,475	0.18 (10/55)	0.56 (14/25)	0.33 (3,088/9,412)	0.42 (6,255/14,983)
Lung	190	127,370	0.13 (1,182/8,820)	0.26 (1,003/3,836)	0.31 (15,359/48,930)	0.42 (27,524/65,784)
Muscle Skeletal	42	19,516	0.21 (16/77)	0.29 (14/48)	0.30 (2,020/6,837)	0.36 (4,572/12,554)
Ovary	140	105,088	0.13 (1,476/11,079)	0.23 (1,482/6,381)	0.30 (10,895/36,923)	0.40 (20,207/50,705)
Prostate	105	88,689	0.11 (388/3,421)	0.26 (394/1,513)	0.31 (10,751/34,421)	0.42 (20,708/49,334)
Testis	47	18,767	0.09 (86/962)	0.30 (96/323)	0.33 (2,123/6,495)	0.41 (4,475/10,987)
Whole Blood	47	52,558	0.09 (70/767)	0.29 (60/205)	0.34 (6,762/19,717)	0.47 (15,118/31,869)

Table R2: Replication rates in the GoDMC data for SNP-CpG associations identified by X-ING (PP>0.8). Proportions and numbers of SNP-CpG pairs identified in GTEx that are also significant in GoDMC ($P < 6 \times 10^{-7}$) are shown. Those SNP-CpG pairs were divided into four groups based on the number of tissues identified and the presence of associations with cis-genes.

References

1. Oliva, M. et al. DNA methylation QTL mapping across diverse human tissues provides molecular links between genetic variation and complex traits. *Nat. Genet.* **55**, 112–122 (2023).
2. Pierce, B. L. et al. Co-occurring expression and methylation QTLs allow detection of common causal variants and shared biological mechanisms. *Nat. Commun.* **9**, 1–12 (2018).
3. Buja, A. & Eyuboglu, N. Remarks on parallel analysis. *Multivariate Behav. Res.* **27**, 509–540 (1992).
4. Franklin, S. B., Gibson, D. J., Robertson, P. A., Pohlmann, J. T. & Fralish, J. S. Parallel analysis: a method for determining significant principal components. *J. Veg. Sci.* **6**, 99–106 (1995).
5. Gölbaşı Şimşek, G. & Aydoğdu, S. Parallel analysis approach for determining dimensionality in canonical correlation analysis. *J. Stat. Comput. Simul.* **86**, 3419–3431 (2016).
6. Li, G., Shabalin, A. A., Rusyn, I., Wright, F. A. & Nobel, A. B. An empirical Bayes approach for multiple tissue eQTL analysis. *Biostatistics* **19**, 391–406 (2018).
7. Liu, J. et al. LLR: a latent low-rank approach to colocalizing genetic risk variants in multiple GWAS. *Bioinformatics* **33**, 3878–3886 (2017).
8. Urbut, S. M., Wang, G., Carbonetto, P. & Stephens, M. Flexible statistical methods for estimating and testing effects in genomic studies with multiple conditions. *Nat. Genet.* **51**, 187–195 (2019).
9. Storey, J. D., Akey, J. M. & Kruglyak, L. Multiple locus linkage analysis of genomewide expression in yeast. *PLoS Biol.* **3**, e267 (2005).
10. Chen, L. S., Emmert-Streib, F. & Storey, J. D. Harnessing naturally randomized transcription to infer regulatory relationships among genes. *Genome Biol.* **8**, 1–13 (2007).
11. Griffiths, D. J. & Schroeter, D. F. (2018). *Introduction to quantum mechanics*. Cambridge university press.
12. The GTEx Consortium. The GTEx Consortium atlas of genetic regulatory effects across human tissues. *Science* **369**, 1318–1330 (2020).
13. Lock, E. F., Hoadley, K. A., Marron, J. S. & Nobel, A. B. Joint and individual variation explained (JIVE) for integrated analysis of multiple data types. *Ann. Appl. Stat.* **7**, 523 (2013).
14. Hu, Y. et al. A statistical framework for cross-tissue transcriptome-wide association analysis. *Nat. Genet.* **51**, 568–576 (2019).
15. Rasiwasia, N., Mahajan, D., Mahadevan, V. & Aggarwal, G. (2014). Cluster canonical correlation analysis. In *Artificial intelligence and statistics* pages 823–831. PMLR.
16. Vasudevan, V. & Ramakrishna, M. A hierarchical singular value decomposition algorithm for low rank matrices. *arXiv preprint arXiv:1710.02812* (2017).
17. Ng, B. et al. Cascading epigenomic analysis for identifying disease genes from the regulatory landscape of GWAS variants. *PLoS Genet.* **17**, e1009918 (2021).
18. Gleason, K. J., Yang, F., Pierce, B. L., He, X. & Chen, L. S. Primo: integration of multiple GWAS and omics QTL summary statistics for elucidation of molecular mechanisms of trait-associated SNPs and detection of pleiotropy in complex traits. *Genome Biol.* **21**, 1–24 (2020).

19. Shu, H., Wang, X. & Zhu, H. D-CCA: A decomposition-based canonical correlation analysis for high-dimensional datasets. *J. Am. Stat. Assoc.* **115**, 292–306 (2020).
20. Newman, A. M. et al. Determining cell type abundance and expression from bulk tissues with digital cytometry. *Nat. Biotechnol.* **37**, 773–782 (2019).
21. Zheng, S. C., Breeze, C. E., Beck, S. & Teschendorff, A. E. Identification of differentially methylated cell types in epigenome-wide association studies. *Nat. Methods* **15**, 1059–1066 (2018).
22. Kim-Hellmuth, S. et al. Cell type-specific genetic regulation of gene expression across human tissues. *Science* **369**, eaaz8528 (2020).
23. Mizuno, A. & Okada, Y. Biological characterization of expression quantitative trait loci (eQTLs) showing tissue-specific opposite directional effects. *Eur. J. Hum. Genet.* **27**, 1745–1756 (2019).
24. Han, B. & Eskin, E. Interpreting meta-analyses of genome-wide association studies. *PLoS Genet.* **8**, e1002555 (2012).
25. Mendioroz, M. et al. Telomere length correlates with subtelomeric DNA methylation in long-term mindfulness practitioners. *Sci. Rep.* **10**, 4564 (2020).
26. Grand, R. S. et al. BANP opens chromatin and activates CpG-island-regulated genes. *Nature* **596**, 133–137 (2021).
27. Taylor, D. L. et al. Integrative analysis of gene expression, DNA methylation, physiological traits, and genetic variation in human skeletal muscle. *Proc. Natl Acad. Sci.* **116**, 10883–10888 (2019).
28. McRae, A. F. et al. Identification of 55,000 replicated DNA methylation QTL. *Sci. Rep.* **8**, 17605 (2018).
29. Min, J. L. et al. Genomic and phenotypic insights from an atlas of genetic effects on DNA methylation. *Nat. Genet.* **53**, 1311–1321 (2021).
30. Qi, T. et al. Identifying gene targets for brain-related traits using transcriptomic and methylomic data from blood. *Nat. Commun.* **9**, 2282 (2018).
31. Vösa, U. et al. Large-scale cis-and trans-eQTL analyses identify thousands of genetic loci and polygenic scores that regulate blood gene expression. *Nat. Genet.* **53**, 1300–1310 (2021).

Reviewer #1 (Remarks to the Author):

All my previous comments have been satisfactorily addressed. In particular, the authors have provided sufficient explanations for the selection of numbers of components for PCA and CCA. The simulation has been substantially updated, which looks more reasonable now.

Regarding the updated version, I have no further comments.

Reviewer #2 (Remarks to the Author):

I'd like to extend my appreciation to the authors for their diligent work. In this updated version, the majority of my previous concerns have been addressed. The untapped potential of the low-rank structure, while recognized in domains like signal processing (Robust PCA, Candès2011) and stock prediction (POET, Fan2013), remains largely unexplored in the GWAS field. This oversight underscores the significance I attribute to X-ING.

However, I'd like to offer some suggestions concerning the presentation in the article:

1. X-ING is supposedly a method designed to efficiently identify which xQTLs are non-zero, capitalizing on the inherent low-rank structure of xQTL data and the correlations across various tissues and omics. Yet, the authors frequently resort to a broader term, "association analysis," throughout the manuscript. This choice of wording might inadvertently lead to confusion. I recommend that the authors more explicitly define the specific problem X-ING seeks to tackle, especially in key sections like the introduction, and the initial part of the results.

2. The practical application of X-ING warrants further detail. I recognize that, upon identifying the non-zero xQTLs with X-ING, enrichment analysis becomes feasible. Still, the current portrayal of this application in the article is somewhat vague. Moreover, I feel that the SCZ section would benefit from a clearer emphasis on the objectives of the analysis and the ways X-ING can facilitate achieving them, rather than delving too deeply into technical intricacies.

Regarding the article's practical implications, I have an inquiry:

3. Joint xQTL and GWAS analysis currently stands as a trending research avenue in statistical genetics. I'm curious if X-ING can enhance joint xQTL and GWAS analysis, especially in contexts like colocalization, TWAS, or Mendelian randomization. I observed that the authors underscored in their discussion that X-ING isn't a novel colocalization method. With that in mind, could X-ING potentially optimize the efficiency of the aforementioned methods or address specific challenges inherent to them?

Pertaining to the statistical inferences, I have some questions:

4. The X-ING model incorporates two random variables: \tilde{z} and γ , with γ being a binary discrete variable. I'm keen to understand the distribution of $q(\tilde{z}, \gamma)$ as presented in equation S2. Is there a closed-form expression for it?

5. My attention was drawn to line 242 on page 9, where the authors interpret the eigenvectors as surrogate variables for tissue-tissue dependence. My conjecture is that these eigenvectors might be output from the CCA analysis, which should be irrelevant to posterior probability and posterior mean. Hence, I advocate for the authors to elucidate in the Method section, using statistical equations, the specific outputs of X-ING and the inferences they facilitate.

Candès E J, Li X, Ma Y, et al. Robust principal component analysis[J]. Journal of the ACM (JACM), 2011, 58(3): 1-37.

Fan J, Liao Y, Mincheva M. Large covariance estimation by thresholding principal orthogonal complements[J]. Journal of the Royal Statistical Society Series B: Statistical Methodology, 2013, 75(4): 603-680.

Reviewer #3 (Remarks to the Author):

I appreciate the authors' efforts to address my previous comments and clarify some key technical issues. My main concerns remain for original comments 1 and 3.

For comment 1, I appreciate that the authors carefully distinguish the concepts of colocalization and "co-occurring." However, I fail to understand the scientific meaning of "co-occurring associations." For example, what is the biological significance of two distinct causal QTLs that are spatially close? And why do we want to analyze multi-tissue, multi-omics data to identify such signals jointly? I can be convinced if the authors provide motivating examples to address these questions.

For comment 3, I also find the authors' notion of labeling opposite-effect QTLs across multiple tissues as "sharing the presence of nonzero effect" misleading. Suppose the authors set out to distinguish zero vs. nonzero effects across tissues. In that case, they definitely should distinguish opposite-direction effects because both represent some qualitative difference in association patterns, and the latter can be argued to have a higher degree of qualitative difference. Logically, I am not convinced by this response.

Reviewer #1 (Remarks to the Author):

All my previous comments have been satisfactorily addressed. In particular, the authors have provided sufficient explanations for the selection of numbers of components for PCA and CCA. The simulation has been substantially updated, which looks more reasonable now.

Regarding the updated version, I have no further comments.

Reviewer #2 (Remarks to the Author):

I'd like to extend my appreciation to the authors for their diligent work. In this updated version, the majority of my previous concerns have been addressed. The untapped potential of the low-rank structure, while recognized in domains like signal processing (Robust PCA, Candès2011) and stock prediction (POET, Fan2013), remains largely unexplored in the GWAS field. This oversight underscores the significance I attribute to X-ING.

However, I'd like to offer some suggestions concerning the presentation in the article:

1. X-ING is supposedly a method designed to efficiently identify which xQTLs are non-zero, capitalizing on the inherent low-rank structure of xQTL data and the correlations across various tissues and omics. Yet, the authors frequently resort to a broader term, "association analysis," throughout the manuscript. This choice of wording might inadvertently lead to confusion. I recommend that the authors more explicitly define the specific problem X-ING seeks to tackle, especially in key sections like the introduction, and the initial part of the results.

Our response #1:

Following the reviewer's suggestion, we replaced the broad term "association analysis" with specific problems X-ING seeks to solve.

1) In the Abstract, we removed the term "association analysis" to avoid confusion and added the following sentences to introduce the motivation of X-ING, "Motivated by the multi-tissue methylation quantitative trait loci (mQTLs) and expression QTLs (eQTLs) analysis, we propose X-ING (Cross-INtegrative Genomics) for cross-omics and cross-context integrative analysis." See page 1.

2) In the Introduction, we removed the term "association analysis" and added the following sentences to discuss the specific problems X-ING seeks to tackle in each application: "In this work, X-ING is used to enhance the power for detecting mQTLs by integrating eQTL statistics. Furthermore, we apply X-ING to examine multi-tissue trans-association effects. Our analysis reveals that associations identified by X-ING are enriched in many known disease/trait-relevant tissue types. Additionally, we illustrate how X-ING can integrate spatial differential expression statistics from spatial transcriptomic studies with multi-tissue eQTL statistics from GTEx. The integrative analysis provides new insights into spatially defined molecular mechanisms underlying diseases." See page 3.

3) In the Results, we change the name of the method overview to "An overview of the X-ING method for cross-omics and cross-context integrative analysis of QTL statistics". See page 4.

2. The practical application of X-ING warrants further detail. I recognize that, upon identifying the non-zero xQTLs with X-ING, enrichment analysis becomes feasible. Still, the current portrayal of this application in the article is somewhat vague. Moreover, I feel that the SCZ section would benefit from a clearer emphasis on the objectives of the analysis and the ways X-ING can facilitate achieving them, rather than delving too deeply into technical intricacies.

Our response #2:

Thanks for the suggestion. The goal of the analysis is to identify genes in cis-association with SCZ risk loci and also show laminar-specific expression (i.e., differential expression across different brain layers). To accomplish this, X-ING integrates two sets of summary statistics, spatial differential expression statistics from LIBD spatial transcriptomic studies and multi-tissue eQTL statistics from GTEx brain tissues. The integration allows X-ING to identify the genes with laminar-specific variation and in cis-association with SCZ risk loci, accounting for the shared and data-specific patterns of the two sets of summary statistics. Furthermore, the results from X-ING show that genes associated with SCZ have laminar-specific expression that is enriched in certain brain lay-

ers, thereby providing new insights into spatially defined mechanisms underlying SCZ genetics. The SCZ section has been carefully rewritten to clarify these points, emphasizing the goal of the analysis and how X-ING facilitates achieving these objectives. Please see page 14 for details.

Regarding the article's practical implications, I have an inquiry:

3. Joint xQTL and GWAS analysis currently stands as a trending research avenue in statistical genetics. I'm curious if X-ING can enhance joint xQTL and GWAS analysis, especially in contexts like colocalization, TWAS, or Mendelian randomization. I observed that the authors underscored in their discussion that X-ING isn't a novel colocalization method. With that in mind, could X-ING potentially optimize the efficiency of the aforementioned methods or address specific challenges inherent to them?

Our response #3:

We would like to thank the reviewer's insights and concur with his/her understanding of the differences between X-ING and other methods such as colocalization, TWAS, or Mendelian randomization (MR). However, X-ING cannot be directly used to optimize these methods due to the fundamental differences in their respective objectives. While it is conceivable to extend these methods within a generalized X-ING framework, for instance, by modeling latent non-zero effect co-occurring patterns in each type of analysis and leveraging information across different data sources. Such extensions are beyond the scope of our current work and the development requires tailored modeling for each type of analysis and is anticipated to be non-trivial. Therefore, we propose to explore these possibilities in future research projects.

Results derived from X-ING should be interpreted as associations but not causality since potential confounders were not explicitly accounted for in the X-ING analysis. Similar to other QTL analysis methods, as noted by Urbut et al. (2019)¹, X-ING does not differentiate between causal associations and those that arise due to linkage disequilibrium (LD). For example, a SNP might appear as a "significant" eQTL because it is in LD with a nearby causal SNP. Compared to methods that aim to infer causation, such as colocalization and Mendelian randomization (MR), X-ING primarily examines associations across omics data types and tissues/cellular contexts accounting for the shared and omics-specific patterns.

Pertaining to the statistical inferences, I have some questions:

4. The X-ING model incorporates two random variables: tilde z and gamma, with gamma being a binary discrete variable. I'm keen to understand the distribution of q(tilde z, gamma) as presented in equation S2. Is there a closed-form expression for it?

Our response #4:

A closed-form expression for the distribution $q(\tilde{z}_\ell, \gamma_\ell)$ is derived through the mean-field method. This approach assumes that the variational posterior distribution is factorizable as

$$q(\tilde{z}_\ell, \gamma_\ell) = \prod_{i=1}^M \prod_{j=1}^{K_\ell} q(\tilde{z}_{ij,\ell}, \gamma_{ij,\ell}),$$

where

$$q(\tilde{z}_{ij,\ell}, \gamma_{ij,\ell}) = q(\tilde{z}_{ij,\ell} | \gamma_{ij,\ell}) q(\gamma_{ij,\ell}).$$

In supplementary material pages 4-6, we derived the variational posterior distribution, which is obtained by maximizing the lower bound in Eqn. (S1). Specifically, the posterior distribution of $\tilde{z}_{ij,\ell}$ is $\mathcal{N}(\mu_{ij,\ell}, s_{ij,\ell}^2)$ when $\gamma_{ij,\ell} = 1$, and $\mathcal{N}(0, \sigma_{j,\ell}^2)$ when $\gamma_{ij,\ell} = 0$. Thus, the posterior distribu-

tion of q is

$$q(\tilde{\mathbf{z}}_\ell, \boldsymbol{\gamma}_\ell) = \prod_{i=1}^M \prod_{j=1}^{K_\ell} \mathcal{N}(\mu_{ij,\ell}, s_{ij,\ell}^2)^{\gamma_{ij,\ell}} \mathcal{N}(0, \sigma_{j,\ell}^2)^{1-\gamma_{ij,\ell}} \alpha_{ij,\ell}^{\gamma_{ij,\ell}} (1 - \alpha_{ij,\ell})^{1-\gamma_{ij,\ell}},$$

where $\alpha_{ij,\ell}$ is the posterior probability of $\gamma_{ij,\ell} = 1$. Definitions of the variational parameters $\mu_{ij,\ell}$, $s_{ij,\ell}^2$ and $\sigma_{j,\ell}^2$ can be found in Eqn. (S5)-(S6) of supplementary material.

5. My attention was drawn to line 242 on page 9, where the authors interpret the eigenvectors as surrogate variables for tissue-tissue dependence. My conjecture is that these eigenvectors might be output from the CCA analysis, which should be irrelevant to posterior probability and posterior mean. Hence, I advocate for the authors to elucidate in the Method section, using statistical equations, the specific outputs of X-ING and the inferences they facilitate.

Our response #5:

Sorry for the confusion. As shown in our Algorithm S1, in each E-step, we evaluated the posterior probability of each tested unit as

$$\alpha_{ij,\ell} = \frac{1}{1 + \exp(-v_{ij,\ell})}, v_{ij,\ell} = \log \frac{\pi_{j,\ell}}{1 - \pi_{j,\ell}} + \frac{1}{2} \left(\log \frac{s_{ij,\ell}^2}{\sigma_{j,\ell}^2} + \frac{\mu_{ij,\ell}^2}{s_{ij,\ell}^2} \right),$$

where $\mu_{ij,\ell}$, $s_{ij,\ell}^2$ and $\sigma_{j,\ell}^2$ are variational parameters (Eqn. (S5) and (S6)). Then, the modulation latent matrices \mathbf{U}_ℓ^* , $\ell = 1, \dots, L$ can be calculated as $\mathbf{U}_\ell^* = \log(\pi_{ij,\ell} / (1 - \pi_{ij,\ell})) - u_{0j,\ell}$. In each M-step, we sequentially regularized these latent matrices using built-in CCA and PCA, respectively.

X-ING outputs the posterior mean and probability of non-zero for each input statistic. Additionally, it provides the eigenvectors from PCA and the canonical coefficients from CCA at the final iteration, and these outputs may facilitate the interpretations of the major patterns in the data.

To clarify these points, we have made the following changes in the Methods Section. On page 19-20, we added "In each E-step, we evaluated the posterior distribution of latent variables and obtained the variational parameters. In the M-step, we extract the common features shared among matrices of logit-transformed posterior probabilities of latent binary status by performing a CCA and PCA." We also added an explicit description of X-ING outputs on page 21.

Candès E J, Li X, Ma Y, et al. Robust principal component analysis?[J]. Journal of the ACM (JACM), 2011, 58(3): 1-37.

Fan J, Liao Y, Mincheva M. Large covariance estimation by thresholding principal orthogonal complements[J]. Journal of the Royal Statistical Society Series B: Statistical Methodology, 2013, 75(4): 603-680.

Reviewer #3 (Remarks to the Author):

I appreciate the authors' efforts to address my previous comments and clarify some key technical issues. My main concerns remain for original comments 1 and 3.

For comment 1, I appreciate that the authors carefully distinguish the concepts of colocalization and "co-occurring." However, I fail to understand the scientific meaning of "co-occurring associations." For example, what is the biological significance of two distinct causal QTLs that are spatially close? And why do we want to analyze multi-tissue, multi-omics data to identify such signals jointly? I can be convinced if the authors provide motivating examples to address these questions.

Our response #6:

Prior research has discussed the scenario where the co-occurrence of e/mQTLs may not be attributed to a single common causal variant. Instead, it could be due to multiple causal variants that are in linkage disequilibrium with each other². Recent studies^{3,4} have also identified regions of complexity where multiple variants (putatively causal) show distinct colocalization patterns. The primary objective of X-ING is not to elucidate the underlying causal mechanisms in complex genetic regions. Instead, it focuses on identifying genetic variants that are associated with molecular traits across multiple contexts. By doing so, X-ING aims to reduce false positives, enhance the power and interpretability of analyses, and identify variants or complex regions for further investigation. This is particularly relevant for regions that involve multiple causal variants, show associations across multiple tissues or contexts, and demonstrate links to multi-omics traits. Results of X-ING can guide further investigations into their underlying genetic and epigenetic mechanisms^{5,6}.

As an illustrative example, previous literature has reported regions with multiple independent (potentially) causal variants that are spatially close³. For instance, in the *EFEMP2* gene region, co-occurring e/mQTL associations involve two distinct causal variants. Expression of *EFEMP2* and methylation at CpG cg14030993 (in whole blood) are influenced by two distinct causal variants (represented by rs78163657 for *EFEMP2* and rs10791824 for cg14030993), each of which colocalizes with distinct GWAS hit for asthma. Despite being distinct, the close proximity of these two causal variants implies a potential shared biology and combined influence on disease risk.

The *AS3MT* locus is another example. A previous study⁴ identified multiple independent association signals for urinary dimethylarsinic acid (DMA%) in the 10q24.32 region (e.g., rs4919687 and rs12573221). There is evidence for colocalization between *AS3MT* cis-eQTLs and the DMA% signal represented by rs4919687 in 21 tissue types and rs12573221 in aortic artery. Colocalization between DMA% signal rs4919687 and mQTLs for cg17932736, cg02786313 and cg15744005 were observed in colon (transverse) and ovary. This example demonstrates some regions can have multiple cis-e/mQTLs in close proximity that are either shared across or specific to different tissues. Complicated colocalization patterns may exist in those regions. This example emphasizes the importance of integrating multi-tissue eQTL and mQTL data for the identification of complex regions. X-ING can pinpoint those regions with complexity; subsequently, additional characterization of the underlying molecular mechanisms can be conducted.

These examples illustrate how X-ING, through the integration of multi-omics data from various tissue types, effectively captures omics-shared and context-shared association patterns. This enables X-ING to differentiate variants and regions with co-occurrence of associations (possibly due to one or multiple potentially causal variants) from variants and regions that lack such associations.

For comment 3, I also find the authors' notion of labeling opposite-effect QTLs across multiple tissues as "sharing the presence of nonzero effect" misleading. Suppose the authors set out to distinguish zero vs. nonzero effects across tissues. In that case, they definitely should distinguish opposite-direction effects because both represent some qualitative difference in association patterns, and the latter can be argued to have a higher degree of qualitative difference. Logically, I am not convinced by this response.

Our response #7:

Previous studies^{6,7} have provided possible explanations for opposite QTL effects across tissues. One potential mechanism is that the SNP tags a single causal variant and the causal variant is activated differently by tissue-dependent factors. Another possible explanation is that the opposite effects across tissues can be explained by SNPs tagging multiple causal variants (in LD), i.e., different causal variants with tissue-specific effects. Those possible mechanisms imply complicated genetic regulation on gene expression in some regions. Because of the LD structure and different tissue-dependent causal variants, SNP-gene associations in those regions show a shared presence of non-zero effects across tissues but the effects are in opposite directions. Many of these complex regions identified by X-ING could be biologically relevant, as disease-relevant QTLs often show tissue-specific effects⁵⁻⁷. Aligning with our previous response #6, X-ING aims to identify these variants and regions needing further attention and make them distinguishable from those lacking such co-occurring associations. Further analyses such as fine-mapping and functional analyses are needed to better understand the underlying mechanisms.

In summary, X-ING aims to identify associations, pinpointing regions with multiple associations to multi-omics traits in multiple tissues. Associations in those regions may show opposite effects across tissues due to the presence of multiple causal variants or different activation and depressing factors. X-ING is not designed for direct causal analysis; rather, it enhances the power and provides a comprehensive view of these regions. Subsequent analyses and additional examinations are needed to further understand the mechanisms of these regions.

References

1. Urbut, S. M., Wang, G., Carbonetto, P. & Stephens, M. Flexible statistical methods for estimating and testing effects in genomic studies with multiple conditions. *Nat. Genet.* **51**, 187–195 (2019).
2. Pierce, B. L. et al. Co-occurring expression and methylation QTLs allow detection of common causal variants and shared biological mechanisms. *Nat. Commun.* **9**, 1–12 (2018).
3. Oliva, M. et al. DNA methylation QTL mapping across diverse human tissues provides molecular links between genetic variation and complex traits. *Nat. Genet.* **55**, 112–122 (2023).
4. Chernoff, M. B. et al. Sequencing-based fine-mapping and in silico functional characterization of the 10q24.32 arsenic metabolism efficiency locus across multiple arsenic-exposed populations. *PLoS Genet.* **19**, e1010588 (2023).
5. Umans, B. D., Battle, A. & Gilad, Y. Where are the disease-associated eQTLs? *Trends Genet.* **37**, 109–124 (2021).
6. Fu, J. et al. Unraveling the regulatory mechanisms underlying tissue-dependent genetic variation of gene expression. *PLoS Genet.* **8**, e1002431 (2012).
7. Mizuno, A. & Okada, Y. Biological characterization of expression quantitative trait loci (eQTLs) showing tissue-specific opposite directional effects. *Eur. J. Hum. Genet.* **27**, 1745–1756 (2019).

Reviewer #2 (Remarks to the Author):

All my previous comments have been satisfactorily addressed. Regarding the updated version, I have no further comments.

Reviewer #3 (Remarks to the Author):

I don't have additional comments to the authors.